



# AIGD-PFT: The first AI-driven Global Daily gap-free 4 km Phytoplankton Functional Type products from 1998 to 2023

Yuan Zhang[1], Fang Shen[1]*, Renhu Li[1], Mengyu Li[1], Zhaoxin Li[1], Songyu Chen[1], Xuerong Sun[2]

[1] State Key Laboratory of Estuarine and Coastal Research, East China Normal University, Shanghai, China.
[2] Centre for Geography and Environmental Science, Department of Earth and Environmental Science, Faculty of Environment, Science and Economy, University of Exeter, Cornwall, United Kingdom.

*Correspondence to*: Fang Shen (fshen@sklec.ecnu.edu.cn)

**Abstract.** Long time series of spatiotemporally continuous phytoplankton functional type (PFT) products are essential for understanding marine ecosystems, global biogeochemical cycles, and effective marine management. In this study, by
integrating artificial intelligence (AI) technology with multi-source marine big data, we have developed a Spatial–Temporal–Ecological Ensemble model based on Deep Learning (STEE-DL), and then generated the first AI-driven Global Daily gap-free 4 km PFTs product from 1998 to 2023 (AIGD-PFT), significantly enhancing the accuracy and spatiotemporal coverage of quantifying eight major PFTs (i.e., Diatoms, Dinoflagellates, Haptophytes, Pelagophytes, Cryptophytes, Green Algae, Prokaryotes, and Prochlorococcus). The input data encompass physical oceanographic, biogeochemical, spatiotemporal
information, and ocean color data (OC-CCI v6.0) that have been gap-filled using a Discrete Cosine Transform with a Penalized Least Square (DCT-PLS) approach. The STEE-DL model utilizes an ensemble strategy with 100 ResNet models, applying Monte Carlo and bootstrapping methods to estimate optimal PFT values and assess model uncertainty through ensemble means and standard deviations. The model's performance was validated using multiple cross-validation strategies—random, spatial-block, and temporal-block—combined with in-situ data, demonstrating STEE-DL's robustness and generalization capability.
The daily updates and seamless nature of the AIGD-PFT product capture the complex dynamics of coastal regions effectively. Finally, through a comparative analysis using a triple-collocation (TC) approach, the competitive advantages of the AIGD-PFT product over existing products were validated. The AIGD-PFT product not only provides the foundation for detailed analyses of PFT trends, interannual variability, and the impacts of climate change on phytoplankton composition across various temporal and spatial scales, but also has the potential to facilitate precise quantification of marine carbon flux and enhances
the accuracy of biogeochemical models. A video demonstration is available at https://doi.org/10.5446/67366 (Zhang and Shen, 2024a). The complete product dataset (1998-2023) can be freely downloaded at https://doi.org/10.11888/RemoteSen.tpdc.301164 (Zhang and Shen, 2024b).

## 1 Introduction

Marine phytoplankton contribute to approximately half of the earth's primary productivity (Field et al., 1998), driving the
operation of marine ecosystems (Beaugrand et al., 2010). These minute organisms are classified into different phytoplankton



functional types (PFTs), playing a crucial role in global biogeochemical cycles, biodiversity, and climate feedbacks (Le Quéré et al., 2005; Gruber et al., 2019). Comprehensive monitoring and research on the spatiotemporal distribution patterns of PFTs are foundational for understanding marine ecosystems, predicting the impacts of climate change (Kramer et al., 2024; Falkowski, 2012). Particularly, for accurately quantification of global ocean carbon fluxes and the improvement of biogeochemical models (Guidi et al., 2016), long-term, high-resolution PFT data is a scientific priority (Nair et al., 2008). Furthermore, as human reliance on marine resources increases, ensuring the sustainability of fisheries (Chassot et al., 2010), effective management of coastal areas, and safeguarding against the risks posed by harmful algal blooms (Xi et al., 2023) all underscore the value of the diversity data of phytoplankton represented by PFTs (Henson et al., 2021).

For the quantification of global PFTs, many analytical techniques and inversion algorithms have been developed in recent years. Among the field sampling analysis methods for quantifying global phytoplankton community composition from water samples, including optical microscopy (Karlson et al., 2010), flow cytometry (Veldhuis and Kraay, 2000), and recent genomics (Catlett et al., 2020), the separation of phytoplankton diagnostic pigments through High-Performance Liquid Chromatography (HPLC) with the assistance of Diagnostic pigment analysis (DPA) or CHEMTAX (Mackey et al., 1996) algorithms remains the most cost-effective and quality-controlled method to date (Swan et al., 2016). The advent of ocean color satellites has enabled continuous global observation. In situ HPLC pigment data and ocean color satellite data have laid the foundation for the development of remote sensing inversion methods, primarily including abundance-based and spectral-based approaches (Mouw et al., 2017; Bracher et al., 2017). Abundance-based indirect methods use chlorophyll-a (Chl-a) concentration as model input, modelling the statistical relationship between Chl-a concentration and diagnostic pigments to retrieve PFTs globally (Hirata et al., 2011; Uitz et al., 2006). Spectral-based methods directly construct relationships between remote sensing reflectance, or absorption spectra, scattering spectra, and the concentrations of different groups, incorporating spectral transformation strategies (such as Principal Component Analysis (Xi et al., 2020), differential spectra (Bracher et al., 2009), etc.) to improve inversion accuracy (Sun et al., 2022). Considering that marine ecological environmental variables (temperature, nutrients, etc.) shape the distribution of different groups through their impact on phytoplankton growth, physiology, and competition, introducing more marine environmental covariates into ecological approaches (Zhang et al., 2023; Raitsos et al., 2008) has become a current research focus: further introducing other biogeochemical and physical oceanographic data on the basis of ocean color satellite data and integrating advanced machine learning methods like random forests and ensemble learning can significantly enhance the accuracy of PFTs modelling.

Based on the aforementioned approaches, several global PFT products have been developed (Table 1), such as (1) a global seasonal surface marine climatology dataset based on CHEMTAX and a global HPLC dataset (Swan et al., 2016); (2) the OC-PFT product based on abundance (Hirata et al., 2011); (3) the PhytoDOAS product based on phytoplankton differential optical absorption spectroscopy (Bracher et al., 2009) ; (4) the synergistic product SynSenPFT that integrates satellite multispectral information with retrievals based on high-resolution PFT absorption properties derived from hyperspectral satellite



measurements (Losa et al., 2017); (5) the EOF-PFT product based on remote sensing reflectance and the empirical orthogonal functions (EOF) algorithm(Xi et al., 2020), along with its modification, the EOF-SST hybrid algorithm (Xi et al., 2021) which

65    incorporates sea surface temperature (SST). In addition to these remote sensing products, the NASA Ocean Biogeochemical Model (NOBM, https://gmao.gsfc.nasa.gov/reanalysis/MERRA-NOBM/data/data_description.php) has been developed, which coupled circulation and radiative models (Gregg and Casey, 2007).

**Table 1** Summary of Existing Open-Source PFT Products

| Product | Method | Spatial resolution | Time resolution | Reference |
|---|---|---|---|---|
| CHEMTAX-PFT | Application of CHEMTAX to a global climatology of pigment data | 1°×1° global grid points | Seasonal climatology | Swan et al. (2016) |
| OC-PFT | Synoptic relationships between Chl-a and its fractional contribution from PFTs | ~4 km | Daily | Hirata et al. (2011) |
| PhytoDOAS | Differential Optical Absorption Spectroscopy (DOAS) | 0.5˚ | Monthly | Bracher et al. (2009) |
| SynSenPFT | Combine synergistically OC-PFT and PhytoDOAS | ~4 km | Daily | Losa et al. (2017) |
| EOF | Empirical orthogonal functions (EOF), using CMEMS GlobColour merged products | ~4 km | Monthly | Xi et al. (2020) |
| EOF-SST | EOF-SST hybrid algorithm | ~4 km | Monthly | Xi et al. (2021) |
| NOBM | NASA Ocean Biogeochemical Model | 1.25° longitude, 2/3° latitude | Daily, Monthly | Gregg and Casey (2007) |

Despite advancements in current algorithms for retrieval PFTs, significant challenges persist in terms of prediction accuracy,

70    spatial coverage, and spatiotemporal resolution. First, abundance-based methods, which rely on Chl-a remote sensing products and empirical formulas to deduce the composition of various PFTs, are computationally straightforward but suffer from limited accuracy and robustness globally (Bracher et al., 2017). Spectral-based methods encounter challenges because of the spectral resolution limitations of current ocean color satellites, which restrict their ability to detect weak phytoplankton signals in optically complex waters. In such environments, non-algal particulate absorption and significant near-infrared water

75    reflectance can obscure diagnostic pigment absorption, potentially rendering spectral-based methods ineffective (Nair et al., 2008). Another significant limitation is the presence of data gaps due to unfavorable conditions, such as orbital configurations, cloud cover, sunlight contamination, and large sensor viewing angles (Mikelsons and Wang, 2019). For instance, the probability of cloud-free conditions over the global ocean for MODIS is only between 25% and 30% (Liu and Wang, 2018).



Although merging images from different satellite missions (e.g., MODIS, VIIRS, OLCI) into the merged product (such as OC-CCI products (Sathyendranath et al., 2019) and CMEMS GlobColour merged products (Garnesson et al., 2019)) has effectively reduced data gaps, the issue of data loss remains severe. This not only results in numerous voids in PFT products but may also introduce biases in trend analysis, obscuring key signals of environmental change and hindering a comprehensive understanding of marine ecosystem dynamics. Such limitations restrict potential applications in climate change research and marine health monitoring. Monthly averaging of data can mitigate the issue of missing data to some extent. However, this approach may conceal significant short-term ecological changes, such as ocean heat waves (Chauhan et al., 2023) and algal blooms (Sadeghi et al., 2012). Additionally, the absence of data also limits the full utilization of on-site data: due to the incompleteness of remote sensing data, many in-situ data cannot be effectively paired with it. This results in the potential inability of models to fully utilize on-site sampling data for calibration or optimization, thereby wasting expensive sampling resources and possibly diminishing the model's generalization capability (Xi et al., 2020). While biogeochemical models offer a global, spatiotemporally continuous PFT modelling approach, their spatial resolution often lacks the detail necessary to accurately reflect local changes and the dynamic characteristics of marine ecosystems.

In summary, although there have been positive developments, current PFT models and products have an imbalance in accuracy, spatio-temporal resolution, spatial coverage and temporal span when compared to existing requirements, suggesting that there is still room for improvement in terms of practicality. The advent of the ocean big data era, coupled with the rise of artificial intelligence technologies such as machine learning, offers new prospects for overcoming the inherent challenges faced by PFT inversion models that currently rely solely on ocean color satellite data (Zhang et al., 2023). Algorithms for data reconstruction and the integration of multi-source data can effectively bridge the observational gaps caused by clouds or orbital, enhancing data utilization efficiency and the continuity of global phytoplankton community monitoring. Furthermore, the application of machine learning and deep learning technologies has the potential to improve the extraction of useful information from vast oceanic datasets. These technologies, capable of processing and analysing large-scale datasets to identify complex patterns and trends, hold the promise of developing high-precision PFT products.

Here, we propose a novel Spatial–Temporal–Ecological Ensemble model based on deep learning (STEE-DL), designed to produce a long time series PFT product. STEE-DL leverages an ensemble of 100 ResNet models, incorporating inputs from reconstructed missing ocean color data, physical reanalysis, biogeochemical, and spatiotemporal information. Utilizing the STEE-DL model, we have produced the first AI-driven Global Daily gap-free 4 km resolution Phytoplankton Functional Type products (AIGD-PFT), include eight major PFTs (i.e., Diatoms, Dinoflagellates, Haptophytes, Pelagophytes, Cryptophytes, Green Algae, Prokaryotes, and Prochlorococcus) from 1998 to 2023. The STEE-DL model's accuracy has been tested through three types of cross-validation (CV) methods: standard, spatial-block, and temporal-block CV. Moreover, we have performed a comprehensive comparison and validation of the AIGD-PFT against other products using triple collocation analysis.



## 2 Methodology

### 2.1 Overall framework

The structure and function of phytoplankton communities are influenced by numerous environmental factors, such as sunlight, nutrient concentration/supply, temperature, carbon chemistry characteristics, and their fluid dynamic environment. We regard the inversion process of PFTs as a nonlinear mapping ($f_x$) problem, aiming to overcome the limitations of relying solely on bio-optical algorithms for predicting the spatial distribution of phytoplankton. This process integrates environmental predictive factors $p$, including bio-optical properties, biogeochemical parameters, physical conditions, and spatio-temporal factors, as shown in equation (1):

$$PG = f_x(p_{Bio-optical}, p_{Biogeochemical}, p_{Physical}, p_{Spatio-temporal}) \tag{1}$$

Building on the work of Zhang et al. (2023), this study further modifies and constructs a STEE-DL model based on a ResNet ensemble to establish $f_x$. An overview of the proposed approach is shown in Figure 1. It specifically includes: (1) based on the global in-situ HPLC dataset compiled by Zhang et al. (2023), this study has expanded and updated it to increase the quantity and diversity of the in-situ data; (2) to address the issue of missing OC data, we utilized the Discrete Cosine Transform with a Penalized Least Square (DCT-PLS) method to reconstruct the data and fill in the missing pixel values; (3) We have integrated multiple sources of marine environmental data as input variables for the regression model; (4) addressing the complex supervised regression problem encountered in multi-source data processing, we trained an ensemble of 100 ResNet models, named the STEE-DL model, to generate daily PFT products for the period from 1998 to 2023.





**Figure 1** Schematic flow of the methodological approach in this study.

## 2.2 Input Datasets and Preprocessing

We first compiled and integrated in situ data obtained by high-performance liquid chromatography (HPLC), and then collected
predictor data including ocean color, physical oceanography, and ocean biogeochemistry for model training and product generation.

### 2.2.1 HPLC Pigment Data

Building upon the updates presented by Zhang et al. (2023), this study integrates additional, newly available HPLC pigment data collected between 1998 and 2023 (refer to Figure 2 for details). This data was primarily sourced from open-access data
repositories such as SeaBASS (https://seabass.gsfc.nasa.gov/), PANGAEA(https://www.pangaea.de/), the British Oceanographic Data Centre (BODC, https://www.bodc.ac.uk/), the Australian Ocean Data Network (AODN, https://portal.aodn.org.au/), and Google Dataset Search (https://datasetsearch.research.google.com/). This initiative has resulted in the acquisition of further HPLC open-source data, leading to the creation of a new global in-situ HPLC pigment database spanning the years 1998 to 2023. In cases of duplicate samples, whether across spatial or temporal dimensions, the
average of the replicates was calculated. By utilizing an updated Diagnostic Pigment Analysis (DPA) methodology, along with newly adjusted weighting coefficients, we conducted DPA to ascertain in-situ PFT Chl-a concentrations. The adjusted coefficients for DPA were referenced from Alvarado et al. (2022) and Xi et al. (2023), with specifics available at https://doi.pangaea.de/10.1594/PANGAEA.954738. From these global HPLC pigment datasets, we selected 6 long-term observation sites as independent validation data. The locations of these sites are shown in Figure 2.

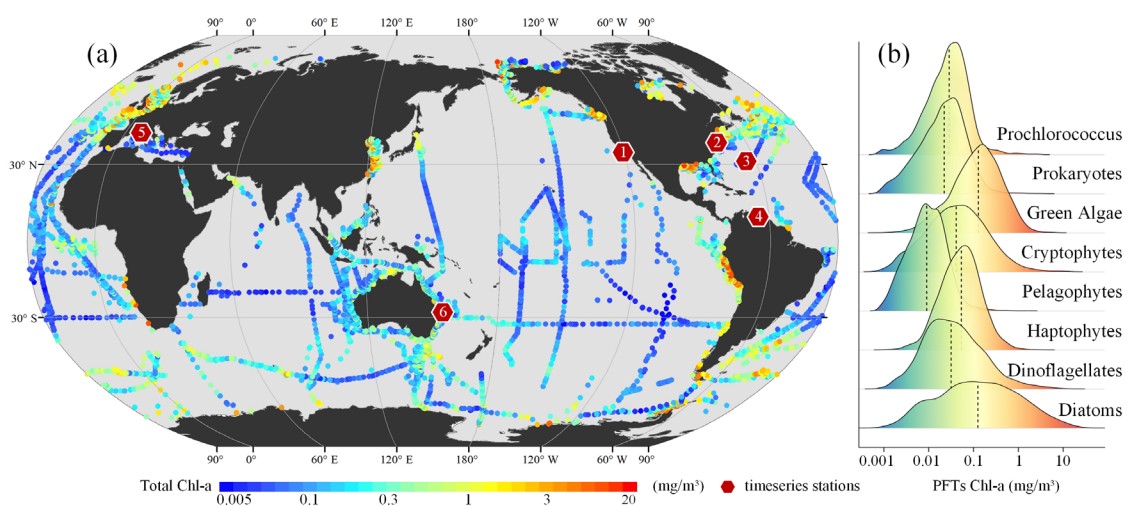


**Figure 2** (a) depicts the spatial distribution of in-situ HPLC pigment datasets, with red hexagons and numbers indicating the locations of six independent long-term time series stations. (b) presents a ridge plot of the probability density distribution for eight types of PFTs.

### 2.2.2 Ocean Color Data and Missing Value Filling

Satellite ocean color remote sensing data is currently the most important data source for the retrieval of PFT. We obtained

daily merged ocean color data from the Ocean-Colour Climate Change Initiative (OC-CCI, version 6.0, https://www.oceancolour.org/) for the period 1998-2023. This data combines measurements from SeaWiFS, MERIS, MODIS-Aqua, and VIIRS sensors and has a spatial resolution of 4 km (Sathyendranath et al., 2019).The raw daily OC-CCI dataset exhibits considerable instances of missing data: Figure 3 illustrates the percentage of valid pixels in the OC-CCI dataset, based on per-pixel statistics spanning the years 1998 to 2023. The results indicate that the majority of marine areas exhibit less than

50% coverage of valid observations, with pronounced gaps particularly evident in higher latitudes.

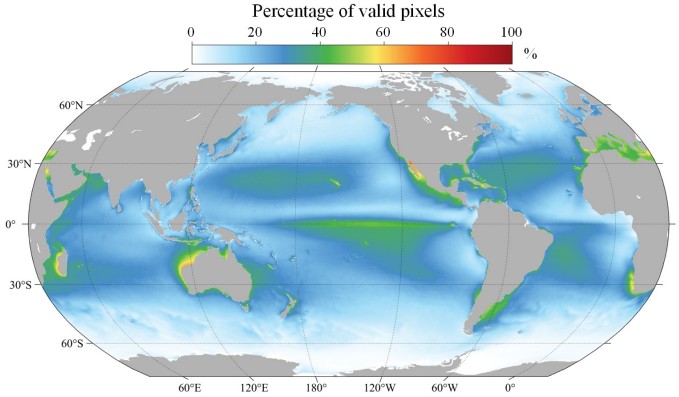

**Figure 3** The percentage of valid pixels in the OC-CCI v6.0 daily dataset.

Given the importance of ocean color data in generating seamless space-time PFT products, it is essential to reprocess missing pixels to fill gaps, thereby maximizing the availability of in-situ and remote sensing data. Previous studies have developed

various methods for reconstructing missing pixels in remote sensing data, such as DINEOF (Data Interpolation Empirical Orthogonal Function) (Liu and Wang, 2022), Optimal Interpolation (Liston and Elder, 2006), and Kriging (Gunes et al., 2006). However, these methods are very time-consuming when dealing with large datasets. For long-term and daily product reconstructions, balancing accuracy and computational efficiency is crucial. Therefore, we adopted the DCT-PLS algorithm, which was initially proposed for automatic smoothing of multidimensional incomplete data (Garcia, 2010). The primary

advantage of the DCT-PLS is its faster speed, while it requires only a small amount of memory storage, and achieves high reconstruction accuracy, making it suitable for processing large datasets. It has been successfully applied to fill data gaps in soil moisture (Wang et al., 2012), NDVI (Yang et al., 2022), coastal ocean surface current (Fredj et al., 2016), and Chl-a (Wang et al., 2022) products. To further improve the computational efficiency of the DCT-PLS algorithm, we modified the original DCT-PLS code, utilizing the built-in FFT computation in PyTorch for GPU-accelerated DCT operations.





Based on the DCT-PLS algorithm, we designed a gap-fill process (as shown in Figure 4), summarized briefly as follows: (1) Data preparation: The original ocean satellite data (e.g., OC-CCI remote sensing reflectance $R_{rs}$, Chl-a concentration, and diffuse attenuation coefficient $K_d490$) are stored in a three-dimensional spatiotemporal data cube. To avoid seams, we directly input the entire global 30-day data cube, with dimensions of 4320×8640×30, representing spatial resolution and a 30-day date-time span, without using regional segmentation. (2) Normalization: To minimize differences in dimensions and magnitudes of

data across different spatial regions, the dataset is standardized by dividing by the spatial mean. (3) DCT-PLS completion: The DCT-PLS method is used to fill in missing values for the target day. We modified Garcia (2010)'s original code (https://www.mathworks.com/matlabcentral/fileexchange/27994-inpaint-over-missing-data-in-1-d-2-d-3-d-nd-arrays?s_tid=prof_contriblnk) to a GPU-accelerated form, significantly improving speed compared to the Matlab-based original code. The entire 30-day time series data undergo a hundred iteration cycles in the DCT-PLS process to fill in the

missing values for the target date. (4) Rolling filling: To enhance the robustness of the filling effect, we adopt a rolling filling strategy. Specifically, for each target day, a 30-day time window is progressively moved forward day by day until the data window moves past that day. This process is repeated 30 times for each target day, with the average of these fillings taken as the final result for the target day. (5) Long time series filling: Following the process described, the entire dataset is traversed and filled day by day, ultimately resulting in a daily continuous and spatially complete data cube from 1998-2023.

This method effectively utilizes time series information to estimate missing values while avoiding discontinuities that might be introduced by data segmentation. Through iteration and averaging, it further improves the accuracy and stability of the filled data. Additionally, through GPU acceleration, this method achieves higher efficiency compared to traditional methods (such as DINEOF). It is important to note that in areas of high latitude with extremely high missing values (exceeding 80%), these data will be directly removed (as demonstrated in the video example available at https://doi.org/10.5446/67366), because

reconstruction under such conditions is impractical.

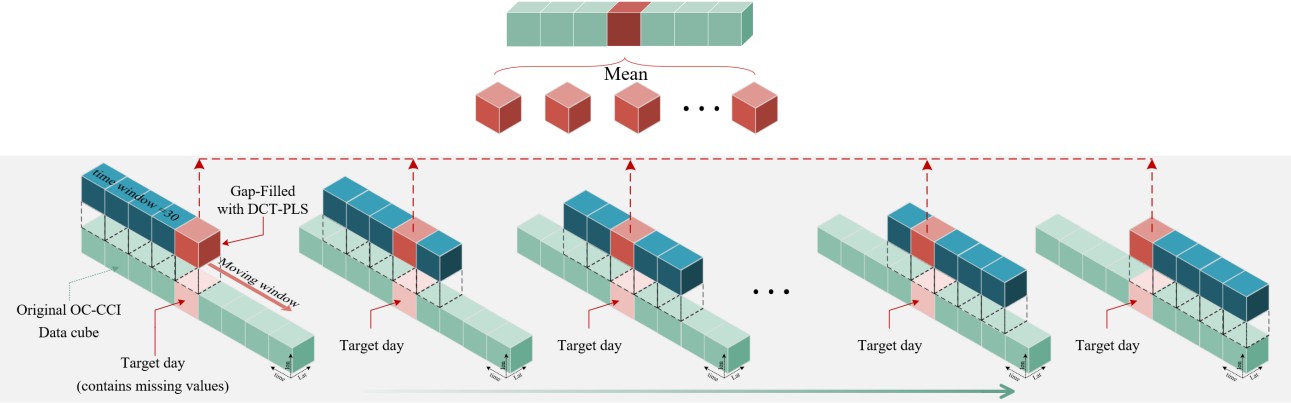

**Figure 4** Gap-fill process with DCT-PLS algorithm.



### 2.2.3 Ocean Physics, Biogeochemistry data, and spatio-temporal information

Incorporation of physical oceanographic data, including Sea Surface Temperature (SST) and Sea Surface Salinity (SSS),
alongside biogeochemical data (Table 2). was performed. These data are provided by the Copernicus Marine Data Store
(https://data.marine.copernicus.eu/products). The SST data are sourced from the ESA SST CCI (Climate Change Initiative)
and C3S (Copernicus Climate Change Service) global Sea Surface Temperature Reprocessed product
(https://doi.org/10.48670/moi-00169). The SSS data are obtained from The Operational Mercator global ocean analysis and
forecast system (https://doi.org/10.48670/moi-00016). Biogeochemical data include nitrate concentration (NC), phosphate
concentration (PC), silicate concentration (SC), and dissolved oxygen (DO). These variables are critical for understanding the
nutrient dynamics in marine ecosystems, which are fundamental factors influencing phytoplankton growth and distribution.
The data for these biogeochemical variables are sourced from the global biogeochemical multi-year hindcast products
(https://doi.org/10.48670/moi-00019). All data undergo the following preprocessing steps: (1) resampling, where all data is
resampled to a 4km resolution using the pysample library (https://doi.org/10.5281/zenodo.3372769). This resampling process
may lead to missing pixels, which are then filled using the nearest neighbor method; (2) standardization: For Rrs, L2 norm
normalization is performed, meaning each band (i.e., $R_{rs412}$, $R_{rs443}$, $R_{rs490}$, $R_{rs510}$, $R_{rs560}$, $R_{rs665}$) is divided by the square root of
the sum of squares of all bands. For Chl-a and $K_d490$, as well as NC, PC, SC, DO, SST, and SSS, standardization is carried
out using the "StandardScaler" function from the scikit-learn library (https://scikit-learn.org/).

Spatial and temporal components were quantified using polar coordinates to facilitate the capture of complex environmental
changes. The spatial term is characterized in Euclidean space using three spherical coordinates $[S_1, S_2, S_3]$ to reflect
autocorrelation and spatial differences. These coordinates represent a point's position in three-dimensional space, calculated
as follows: (1) $S_2$ describes the component in the east-west direction, calculated by longitude, with the formula $S_1 = \sin\left(2\pi \frac{lon}{360}\right)$; (2) $S_2$ combines longitude and latitude to provide the position in the north-south direction and the vertical
distance from the equator, calculated as $S_2 = \cos\left(2\pi \frac{lon}{360}\right) \sin\left(2\pi \frac{lat}{180}\right)$; (3) $S_3$ represents the straight-line distance from the
center of the Earth to the point, calculated as $S_3 = \cos\left(2\pi \frac{lon}{360}\right) \cos\left(2\pi \frac{lat}{180}\right)$. Furthermore, the temporal term ($T \sim [T_1, T_2]$) is
represented by two sine and cosine functions of the day of the year ($DOY$), enabling the capture of both daily variations and
seasonal patterns of PFT. Here, $T_1 = \cos\left(2\pi \cdot \frac{DOY}{N_{day}}\right)$ and $T_2 = \sin\left(2\pi \cdot \frac{DOY}{N_{day}}\right)$, where $N_{day}$ is the total number of days in the
corresponding year.






**Table 2** Predictors and corresponding data products.

| Dataset | Abbreviation | Definition | Resolution |
|---|---|---|---|
| Ocean color data | $R_{rs412\text{-}670}$ | Remote sensing reflectance at 412, 443, 490, 510,555 and 670 nm | ~4 km, Daily, 1998.1.1-2023.12.31 |
| | $K_d490$ | diffuse attenuation coefficient at 490 nm | |
| | Chl-a | Chlorophyll-a concentration | |
| Biogeochemistry data | NC | Nitrate concentration | 1/4 °, Daily, 1998.1.1-2023.12.31 |
| | PC | Phosphate concentration | |
| | SC | Silicate concentration | |
| | DO | Dissolved oxygen | |
| Ocean Physical data | SST | sea surface temperature | 1/20°, Daily, 1998.1.1-2023.12.31 |
| | SSS | sea surface salinity | 1/12°, Daily, 1998.1.1-2023.12.31 |
| Spatio-temporal information | $S_1$ | $S_1 = \sin\left(2\pi \frac{lon}{360}\right)$ | – |
| | $S_2$ | $S_2 = \cos\left(2\pi \frac{lon}{360}\right)\sin\left(2\pi \frac{lat}{180}\right)$ | |
| | $S_3$ | $S_3 = \cos\left(2\pi \frac{lon}{360}\right)\cos\left(2\pi \frac{lat}{180}\right)$ | |
| | $T_1$ | $T_1 = \cos\left(2\pi \cdot \frac{DOY}{N_{day}}\right)$ | |
| | $T_2$ | $T_2 = \sin\left(2\pi \cdot \frac{DOY}{N_{day}}\right)$ | |



### 2.3 Spatial–Temporal–Ecological Ensemble model based on deep learning

#### 2.3.1 Network Architecture

Ensemble learning has emerged as a powerful approach to enhancing prediction performance by combining the outputs of multiple models. STEE-DL Models that use deep ensemble learning combine the advantages of deep learning with those of ensemble learning to achieve better generalization. STEE-DL model framework introduces an ensemble consisting of N residual neural networks (ResNet) as its components. The ResNet is known for their shortcut connections, which help in maintaining a smooth flow of gradients during the learning process. To ensure efficiency, each component model is built with

two residual blocks designed to reduce computational demands while preserving the effectiveness of a deep network. These blocks comprise a fully connected layer, a ReLU activation function, and a shortcut connection for uninterrupted information transmission. This setup decreases the dimensionality of features from 19 to 16, and then to 10, before a final fully connected layer maps these features to an output value for predicting the target variable. Research, such as the work by Gen and colleagues, has shown that ensemble stability improves significantly when the number of component models, N, exceeds 50, but the

marginal gains in reducing standard error diminish after reaching 100 models. Therefore, aiming for a balance between accuracy and computational efficiency, we have chosen an ensemble size of N=100. Based on this architecture, we have implemented the STEE-DL models using PyTorch (https://pytorch.org/).

#### 2.3.2 Model Ensemble and Uncertainty

Each ResNet within the ensemble focuses on different subsets and features of the training data, The mean (μ) of the outputs

from the 100 independent models is considered the optimal estimation of the target variable

$$\mu_{pft} = \sum_{i=1}^{i=100} \text{PFT}_{estimated(i)} \Big/ 100 \tag{2}$$

The variability among ensemble model outputs, quantified by the standard deviation ($\sigma$) of the 100 independent models, provides a measure of uncertainty in predictions. This uncertainty reflects the variability in predictions due to differences in training sets, initializations, and learning dynamics. A higher standard deviation indicates greater disagreement among models, suggesting lower confidence in the prediction.

$$\sigma = \sqrt{\sum_{i=1}^{i=100} \left( \text{PFT}_{estimated(i)} - \mu_{pft} \right)^2 \Big/ 100} \tag{3}$$

this approach differs from statistical methods based on error propagation, which evaluate prediction uncertainty by analyzing input data uncertainties (e.g., measurement errors) and their transmission through the model to the outputs. Such methods

require a clear understanding of input error distributions and typically assume these errors are independent. Given the STEE-DL model's reliance on diverse marine and in situ High-Performance Liquid Chromatography (HPLC) data of varying quality control, accurately applying error propagation for uncertainty measurement is challenging. Our ensemble-based approach primarily addresses model uncertainty but also indirectly reveals data uncertainties by demonstrating how predictions respond to variations in representation and data subsets.

### 2.3.3 Training Procedure

To compile the training dataset, we align in-situ HPLC data with reconstructed OC-CCI and environmental data, both spatially and temporally. This alignment projects the data onto a 4km grid according to the latitude, longitude, and date of the HPLC measurements. In cases where several HPLC measurements are located within the same 4km grid cell, we average these measurements to consolidate corresponding predictor variables.

The STEE-DL model utilizes a Monte Carlo and bootstrapping ensemble learning approach to boost model stability and predictive accuracy. By resampling, it randomly selects two-thirds of the total dataset as the training set for each iteration, repeating this procedure 100 times. This method is designed to create a varied collection of models by multiple rounds of sampling, significantly improving the model's ability to generalize. This reduces the model's reliance on specific data distributions, thereby increasing both the accuracy and the robustness of its predictions.

Throughout the training phase, the model optimization relies on the Adam optimizer, complemented by L1 regularization to promote sparsity within the model and prevent overfitting. Gradient clipping is applied to manage potential issues with exploding gradients, thus ensuring a more stable training process. An Exponential Moving Average (EMA) strategy is employed to stabilize the model weights by averaging them over time, which helps to minimize variations and secure a consistent performance from the final model.

To circumvent the issue of the model predicting unreasonably high values during training, we have crafted a specialized loss function. This function incorporates the traditional Mean Squared Error (MSE) while imposing extra penalties on predictions that surpass set thresholds. Not only does this effectively prevent the model from making unrealistic predictions, but it also guides the model towards more accurate parameter adjustments, assuring that its predictions stay within feasible limits.

### 2.4 Evaluation strategies

To comprehensively test the accuracy and robustness of the model, the evaluation of the STEE-DL model comprises two parts: first, the model performance is validated using a five-fold cross-validation method in three different ways; second, the evaluation is based on a tripartite matching analysis algorithm.





### 2.4.1 Cross-validation Approach


Cross validation (CV) is a commonly used method for analyzing model performance, allowing for a comprehensive assessment of a model's accuracy, stability, and generalization. This study implements three types of CV methods: random five-fold CV, time-block five-fold CV, and spatial-block five-fold CV, to deeply evaluate the model's multifaceted performance. Standard five-fold cross-validation: This method randomly divides all data into five equal-sized subsets. In each round of validation,

one subset is selected as the test set, while the remaining four subsets serve as the training set, ensuring that each data point is used as test data. This method primarily evaluates the model's performance and generalization on the entire dataset. Time-block five-fold cross-validation: Data is divided into five consecutive time periods in chronological order. In each iteration, data from one time period is chosen as the test set, with the data from the remaining periods serving as the training set (as shown in Figure 5). This method takes into account the continuity and dependency of time series, helping to evaluate the

model's ability to capture time trends and seasonal variations.



**Figure 5** Temporal block CV procedure.

Spatial-block five-fold cross-validation: Similar to time-block cross-validation, but data is divided based on spatial location. A hexagonal grid was created at 20° horizontal and vertical intervals, and regions without sampling points were removed for

hexagonal regions. In each round, data from one geographical block is left out as the testset, while data from other blocks are used for training(as shown in Figure 6). This method prevents potential data leakage due to spatial autocorrelation and helps to assess the model's spatial prediction capability and its generalization across different geographical locations.

The coefficient of determination ($R^2$), root mean square error (RMSE), mean absolute error (MAE), and symmetric mean absolute percentage error (sMAPE) were utilized to quantify the performance of the model, according to:

$$R^2 = 1 - \frac{\sum_{i=1}^{N}[p_i - \hat{p}_i]^2}{\sum_{i=1}^{n}[p_i - \bar{p}]^2} \tag{4}$$





$$\text{RMSE} = \left[ \frac{1}{N} \sum_{i=1}^{N} (p_i - \hat{p}_i)^2 \right]^{1/2} \tag{5}$$

$$\text{MAE} = \frac{1}{N} \sum_{i=1}^{N} |p_i - \hat{p}_i| \tag{6}$$

$$\text{sMAPE} = \frac{100}{N} \sum_{i=1}^{N} \frac{|\hat{p}_i - p_i|}{(\hat{p}_i - p_i)/2} \tag{7}$$

where $p_i$ and $\hat{p}_i$ are the log10-scaled observed and estimated of each PFT for sample $i$, $N$ is the number of observations, $\bar{p}$ is the log10-scaled mean of the observed values.

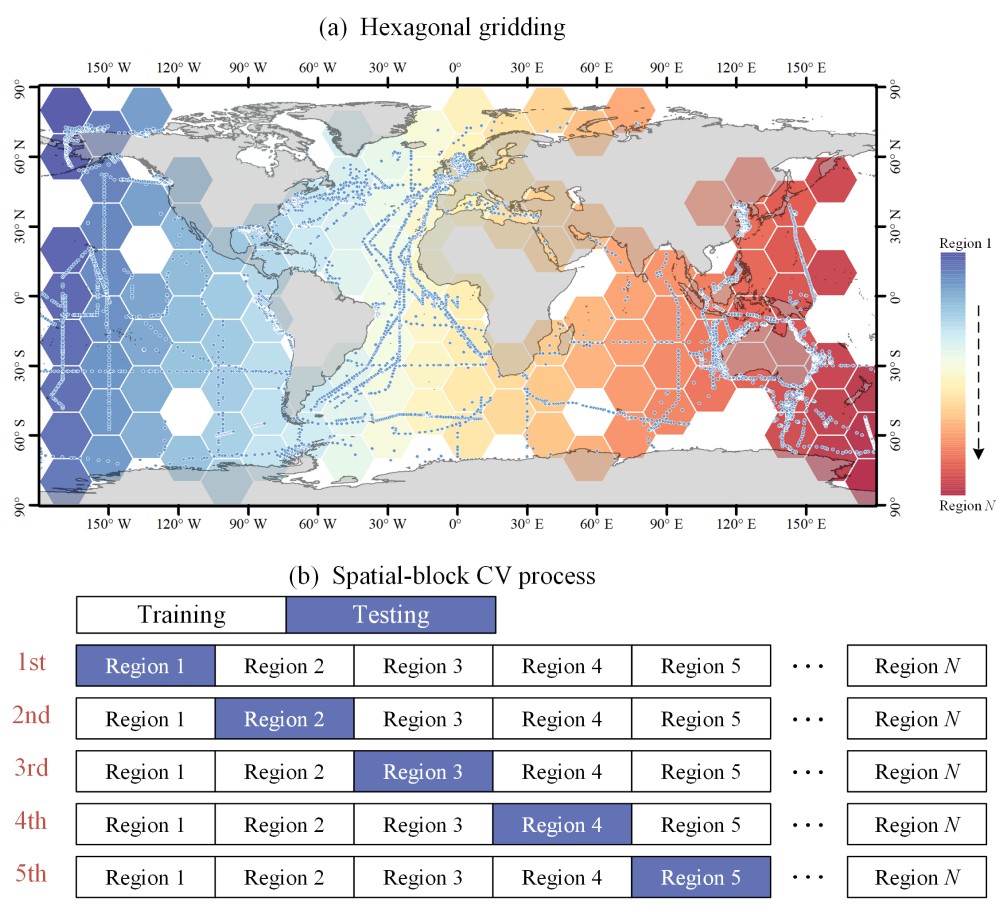

**Figure 6** Spatial block CV procedure.



### 2.4.2 Triple Collocation Analysis

The Triple Collocation Analysis (TCA) method was also utilized for a global evaluation of the AIGD-PFT product. TCA is a technique that allows for the assessment and quantification of error characteristics in three independent data sources without relying on reference data pre-assumed to be "true"(Mccoll et al., 2014). This method has been widely adopted in the uncertainty evaluation of remote sensing products across various fields, including soil moisture (Kim et al., 2023), sea surface salinity (Hoareau et al., 2018), and sea surface temperature (Saleh and Al-Anzi, 2021).

For error statistics based on TCA, we selected the fractional Mean Squared Error (*fMSE*) and the squared correlation coefficient. These metrics offer direct insights into data precision and accuracy. *fMSE*, in particular, is beneficial because it quantifies the relative error in a product, scaling from 0 to 1, where a lower value indicates higher precision. *fMSE* calculated as follows:

$$fMSE_i = \frac{\sigma_{\varepsilon_i}^2}{\sigma_i^2} = \frac{\sigma_{\varepsilon_i}^2}{\beta_i^2 \sigma_\Theta^2 + \sigma_{\varepsilon_i}^2} = \frac{1}{1 + SNR_i} \tag{8}$$

With $i = \alpha_i + \beta_i\Theta + \varepsilon_i$ , corresponds to three spatially and temporally collocated datasets $[X, Y, Z]$. $\sigma_{\varepsilon_i}^2$ is the TCA-based error variance of an individual product. $\beta_i$ and $\alpha_i$ represents the scaling factor and systematic additive biases between the unknown

true signal $\Theta$ and the datasets $i$. $\sigma_i^2$ is the variance of the individual data, $\sigma_\Theta^2$ is the variance of the true signal, and *SNR* is the Signal-to-Noise Ratio. The *fMSE* value below 0.5 suggests that the true signal is a more significant component of the data than the estimation noise, indicating a precise product. Similarly, the squared correlation coefficient ($R_i^2$) is defined as:

$$R_i^2 = \frac{\beta_i^2 \sigma_\Theta^2}{\beta_i^2 \sigma_\Theta^2 + \sigma_{\varepsilon_i}^2} = \frac{SNR_i}{1 + SNR_i} \tag{9}$$

The foundational assumptions of TCA are important for its application (Kim et al., 2023): (1) a linear relationship exists between each dataset and the true signal, (2) the errors among the datasets are orthogonal, and (3) there's no correlation among

the errors of different datasets. These principles ensure the robustness of the TCA method in providing an unbiased error and quality assessment of products.

Several other PFT products were introduced and organized into triads for TCA analysis. First, SynSenPFT (https://doi.org/10.1594/PANGAEA.875873) and NOBM-daily products were obtained, forming a daily product triplets. Both SynSenPFT and NOBM-daily contain three PFTs - diatoms, cyanobacteria (prokaryotes), and coccolithophores (main

contributing PFT to Haptophytes). TCA evaluations were conducted separately for these three PFTs. The TCA calculation process selected overlapping time periods of SynSenPFT, NOBM-daily, and the proposed AIGD-PFT products, from August 1, 2002, to March 31, 2012, totaling 3,515 days. All three products were resampled to a 1° resolution. Similarly, we also obtained EOF-PFT data (https://doi.org/10.48670/moi-00281) and NOBM-monthly product to form a monthly triplets, again

conducting TCA assessments for diatoms, prokaryotes, and Haptophytes. Before evaluation, the AIGD-PFT products were
merged monthly and resampled to 1° resolution along with EOF-PFT and NOBM-monthly. The temporal span of monthly
TCA triplets products was from January 2003 to December 2017, totaling 180 months. NOBM's daily and monthly data are
all obtained from Giovanni website (https://giovanni.gsfc.nasa.gov/). We additionally employed RECCAP2 ocean regions for
regional TCA statistics, as shown in Figure 7.

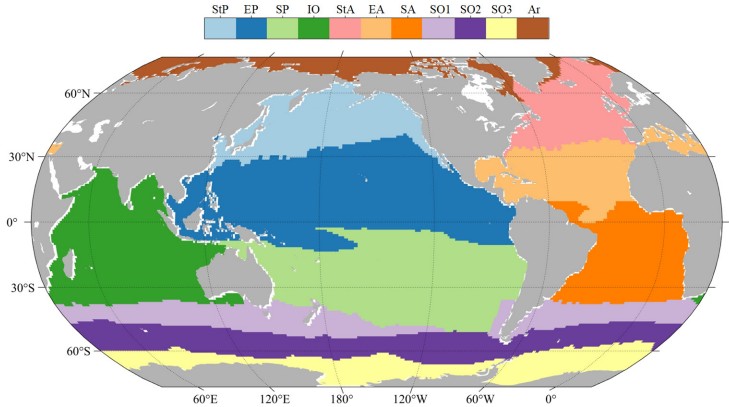

**Figure 7** Map of RECCAP2-ocean regions (Regional Carbon Cycle Assessment and Processes, Canadell et al. (2011), https://reccap2-ocean.github.io/regions/), include Arctic (Ar), Subtropical Atlantic (StA), Equatorial Atlantic (EA) , South Atlantic (SA), Subtropical Pacific (StP) , Equatorial Pacific (EP) , South Pacific (SP), Indian Ocean (IO), Southern Ocean (SO).

## 3 Result

### 3.1 Model verification

### 3.1.1 Three CV Methods

To comprehensively assess the performance of the proposed STEE-DL model, three five-fold cross-validation (CV) methods
were implemented: random, temporal-block, and spatial-block CV. The results are shown in Table 3. The random CV analysis
revealed generally high prediction accuracy across all 8 PFTs, as visualized by the scatter plot in Figure 8. Diatoms exhibited
highest performance, achieving $R^2$ of 0.8. This confirms the STEE-DL model's strong capability in Diatom prediction.
Conversely, Pelagophytes displayed the weakest performance, reflected by a $R^2$ of just 0.5. Further examination through the
probability distribution histograms and Cumulative Distribution Function (CDF) curves of predicted versus actual values
revealed a good alignment, indicating the model's overall ability to accurately mimic observed data distributions. However, a
notable limitation observed was the STEE-DL model's tendency towards overestimating lower values and underestimating
higher values. This suggests a bias towards predicting smoother values, potentially resulting in less accurate predictions for
extreme high or low actual values.

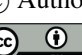



**Table 3** Model performance metrics ($R^2$, MAE, RMSE, and sMAPE, based on random, temporal-block, and spatial-block five-fold CV procedure)

| PFT | Metrics | Cross-validation approach | | |
|---|---|---|---|---|
| | | random CV | temporal-block | spatial-block |
| Diatoms | $R^2$ | 0.86 | 0.82 | 0.81 |
| | MAE | 0.26 | 0.29 | 0.30 |
| | RMSE | 0.33 | 0.37 | 0.40 |
| | sMAPE | 51.21 | 55.53 | 54.25 |
| Dinoflagellates | $R^2$ | 0.71 | 0.62 | 0.64 |
| | MAE | 0.26 | 0.30 | 0.30 |
| | RMSE | 0.33 | 0.39 | 0.40 |
| | sMAPE | 23.91 | 27.16 | 28.75 |
| Haptophytes | $R^2$ | 0.60 | 0.50 | 0.51 |
| | MAE | 0.21 | 0.23 | 0.23 |
| | RMSE | 0.26 | 0.30 | 0.31 |
| | sMAPE | 17.73 | 20.24 | 20.49 |
| Pelagophytes | $R^2$ | 0.50 | 0.39 | 0.42 |
| | MAE | 0.23 | 0.26 | 0.25 |
| | RMSE | 0.29 | 0.33 | 0.34 |
| | sMAPE | 11.45 | 12.83 | 12.55 |
| Cryptophytes | $R^2$ | 0.68 | 0.57 | 0.61 |
| | MAE | 0.29 | 0.34 | 0.33 |
| | RMSE | 0.36 | 0.43 | 0.43 |
| | sMAPE | 26.31 | 30.55 | 29.56 |
| Green algae | $R^2$ | 0.72 | 0.65 | 0.64 |
| | MAE | 0.22 | 0.25 | 0.25 |
| | RMSE | 0.27 | 0.31 | 0.33 |
| | sMAPE | 33.16 | 36.57 | 36.11 |
| Prokaryotes | $R^2$ | 0.68 | 0.59 | 0.59 |
| | MAE | 0.23 | 0.26 | 0.26 |
| | RMSE | 0.28 | 0.33 | 0.34 |
| | sMAPE | 13.82 | 15.76 | 15.78 |
| Prochlorococcus | $R^2$ | 0.55 | 0.19 | 0.32 |
| | MAE | 0.22 | 0.29 | 0.28 |
| | RMSE | 0.28 | 0.40 | 0.41 |
| | sMAPE | 14.71 | 18.37 | 17.06 |

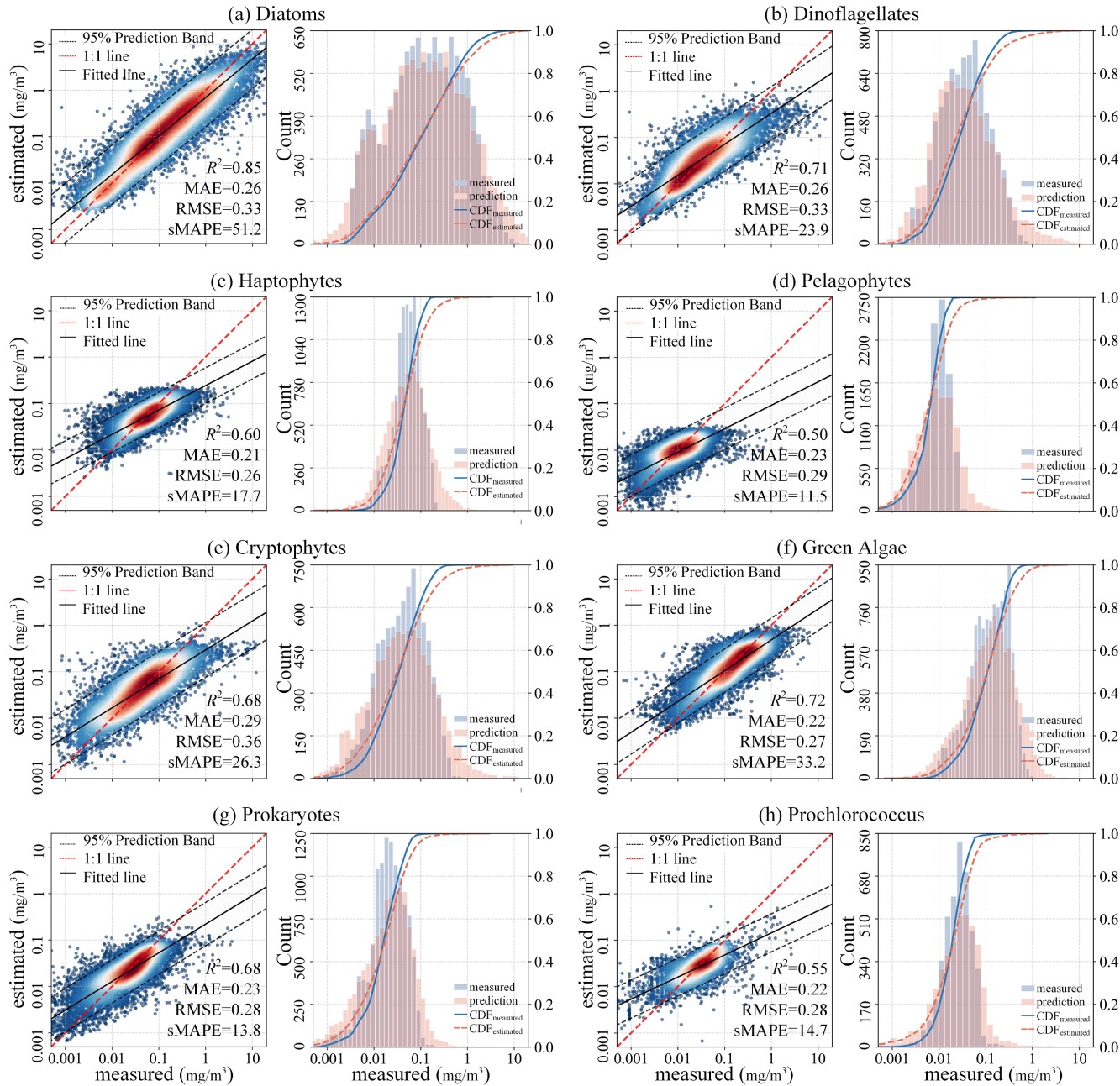

**Figure 8** Scatter diagrams, probability distribution and CDF (based on random five-fold CV procedure) of the predicted vs. measured Chl-a concentrations of 8 PFTs.

By comparing the model performance under three different CV strategies, we delved further into the STEE-DL model's generalization abilities in terms of time and space. Figure 9 reveals that the STEE-DL model's accuracy decreases under

temporal and spatial cross-validation compared to standard random cross-validation. Notably, the predictive accuracy for

diatoms was minimally affected by the different validation strategies, with $R^2$ values remaining above 0.8 for all three methods. This demonstrates the model's robust generalization capability in both temporal and spatial aspects. Except for the Prochlorococcus, the decrease in accuracy was modest for other PFTs in spatial cross-validation (with about a 0.1 decrease in $R^2$ and a 0.5 increase in MAE), suggesting that the STEE-DL model is relatively robust and can accurately estimate regions lacking in situ observational data. Compared to spatial validation, there was a slight decrease in accuracy for temporal cross-

validation, but it still maintained a good level. Except for a significant drop in temporal generalization for the Prochlorococcus, the temporal cross-validation accuracy for other PFTs remained favorable.

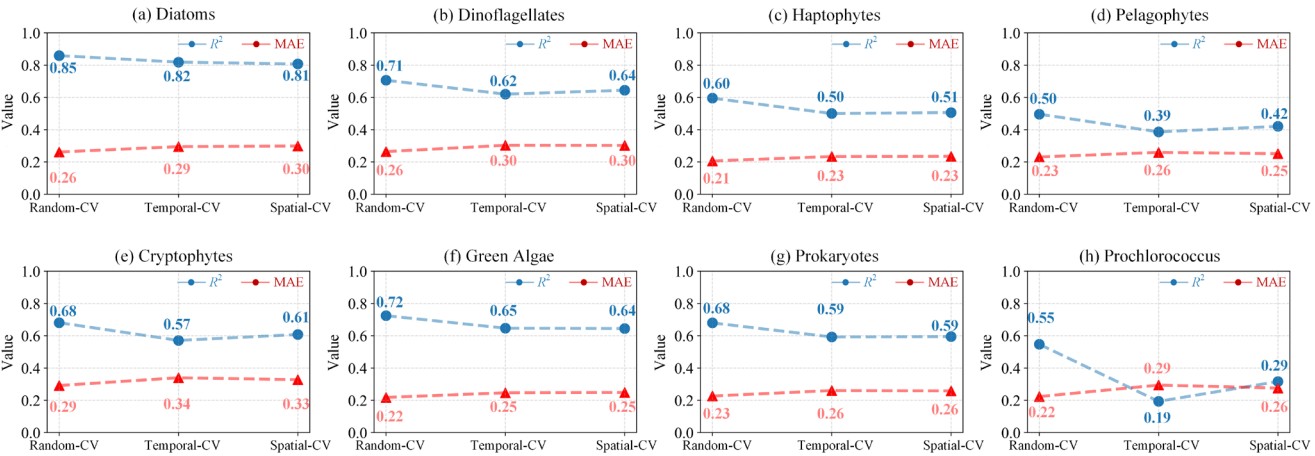

**Figure 9** Comparison of the results obtained using different CV methods, including random CV, spatial block CV, and temporal block CV. Blue indicates variations in the $R^2$ under the three cross-validation methods, while red represents changes in MAE.

During the training process of the STEE-DL model, two types of training data are utilized: "original match" training data and "reconstructed match" training data. The "original match" training data refers to data successfully matched directly from the in situ HPLC database and the OC-CCI original data; the "reconstructed match" training data refers to matched data obtained after completing the missing parts of OC-CCI data using the DCT-PLS technique. By comparing the model's prediction accuracy on these two types of data, we can assess not only the STEE-DL model's adaptability to changes in data completeness

but also verify the effectiveness and accuracy of the DCT-PLS technique in reconstructing missing ocean color data. If the STEE-DL model's performance on the "reconstructed match" data is similar to its performance on the "original match" data, it not only indicates that the DCT-PLS method is effective and reasonable for reconstructing ocean color data, but also confirms that the STEE-DL model can provide reliable PFT predictions under varying data quality and completeness conditions.

We calculated the $R^2$ between predicted and actual values for both original and reconstructed pixels using the three cross-

validation methods (Figure 10). Except for a significant difference in performance for Prochlorococcus, the accuracy of





reconstructed pixels was generally consistent with that of the original pixels, demonstrating good performance. This indicates that the reconstructed pixels did not degrade model performance, thus confirming both the high congruency of our data reconstruction method with actual conditions and the robustness of the STEE-DL model.

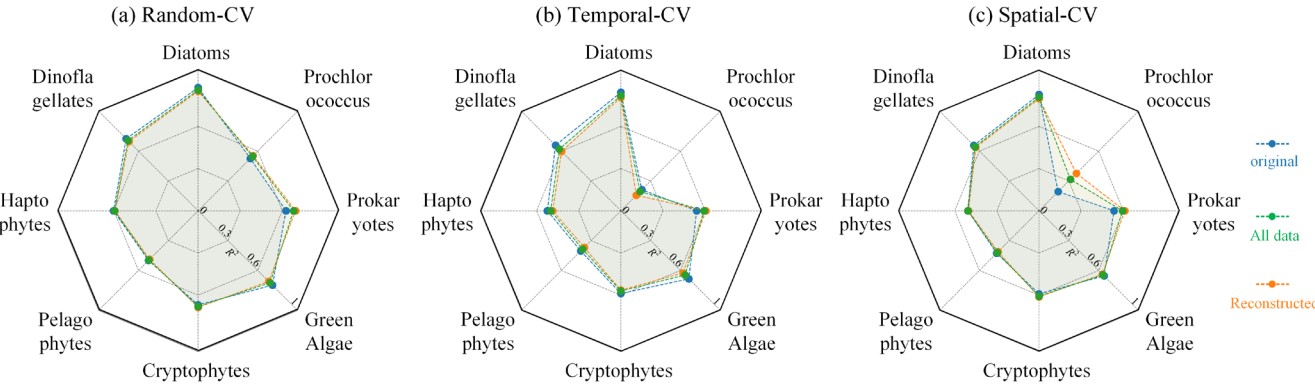

**Figure 10** Model performance comparison on original (blue dashed), reconstructed (orange dashed), and all pixels (orange solid) using (a) random CV, (b) temporal CV, and (c) spatial CV.

### 3.1.2 Long-time Series Observations

The effectiveness of the proposed STEE-DL model was validated using data from six independent long-term observation sites. The results, as shown in the Figure 11, display the correlation coefficients between predicted and actual values at these six sites. The STEE-DL model demonstrated varying degrees of predictive capability across different sites and PFTs. Firstly, the model achieved high prediction accuracy for key ecological types such as Diatoms, Dinoflagellates, and Green algae, with significant advantages at certain sites: for instance, at sites 4 and 5, the prediction correlation coefficients for Diatoms were as high as 0.90 and 0.88, respectively. Sites 5 exhibited high correlations for Dinoflagellates and Green algae predictions, reaching 0.69 and 0.83, respectively, highlighting the model's ability to accurately capture the dynamics of these major ecological types. However, it is noteworthy that predictions for certain ecological types showed considerable fluctuations at specific sites. For example, site 3 had a prediction correlation coefficient of 0.90 for Pelagophytes but a relatively lower coefficient of 0.48 for Dinoflagellates. In terms of ecological types like Prokaryotes and Prochlorococcus, the model's predictions were generally more balanced, with site 2 showing a high correlation coefficient of 0.80 for Prochlorococcus. Overall, despite some fluctuations and differences, these results emphasize the STEE-DL model's capability to capture the temporal trends of different PFTs with relative accuracy.

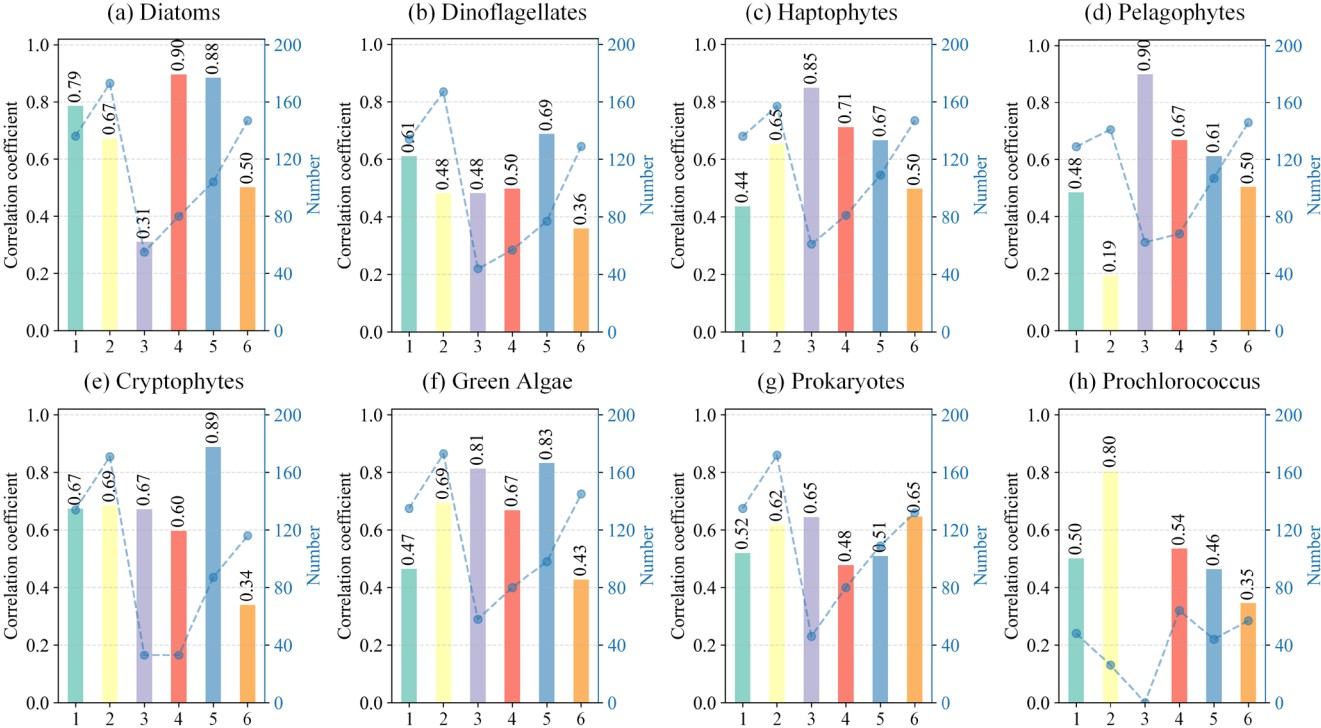

**Figure 11** STEE-DL model performance at six independent time series stations. Correlation coefficient (bar chart) and number of successfully matched pixels (blue dashed line).

## 3.2 Gap-free PFT product and Uncertainties

Following the validation of the STEE-DL model, it was retrained with the entirety of the data available, enabling the generation of a long time series and spatiotemporally continuous AIGD-PFT product for the period from 1998 to 2023. An example from this dataset, depicted in Figure 12 for March 10, 2020, demonstrates the results of the AIGD-PFT. Notably, while nearly half of the original OC-CCI data contained missing values (as shown in Figure 12a), our reconstructed dataset has achieved spatial completeness with good continuity. Within this dataset, the distribution patterns of the eight PFTs showed significant

variability. For example, diatoms were primarily found in the oceanic regions of mid to high latitudes (30°–60°), thriving in nutrient-rich, cold waters, and areas affected by terrestrial runoff. Dinoflagellates, with a distribution pattern similar to diatoms, were mostly present in the nutrient-rich upwelling zones of high latitudes and nearshore areas, though their content was relatively lower. Prokaryotes were noted for maintaining higher concentrations in the nutrient-poor, sunlight-abundant waters of tropical and subtropical regions (0°–30°), with a significant decrease in biomass at higher latitudes, a characteristic closely

resembling that of Prochlorococcus. Haptophytes and green algae were observed more frequently in the subtropical regions of the Pacific, Atlantic, and the Southern Ocean, reaching into mid to high latitudes. In contrast, Pelagophytes and Cryptophytes were found to be more prevalent in tropical and subtropical regions, showing lower concentrations in areas of lower latitude.

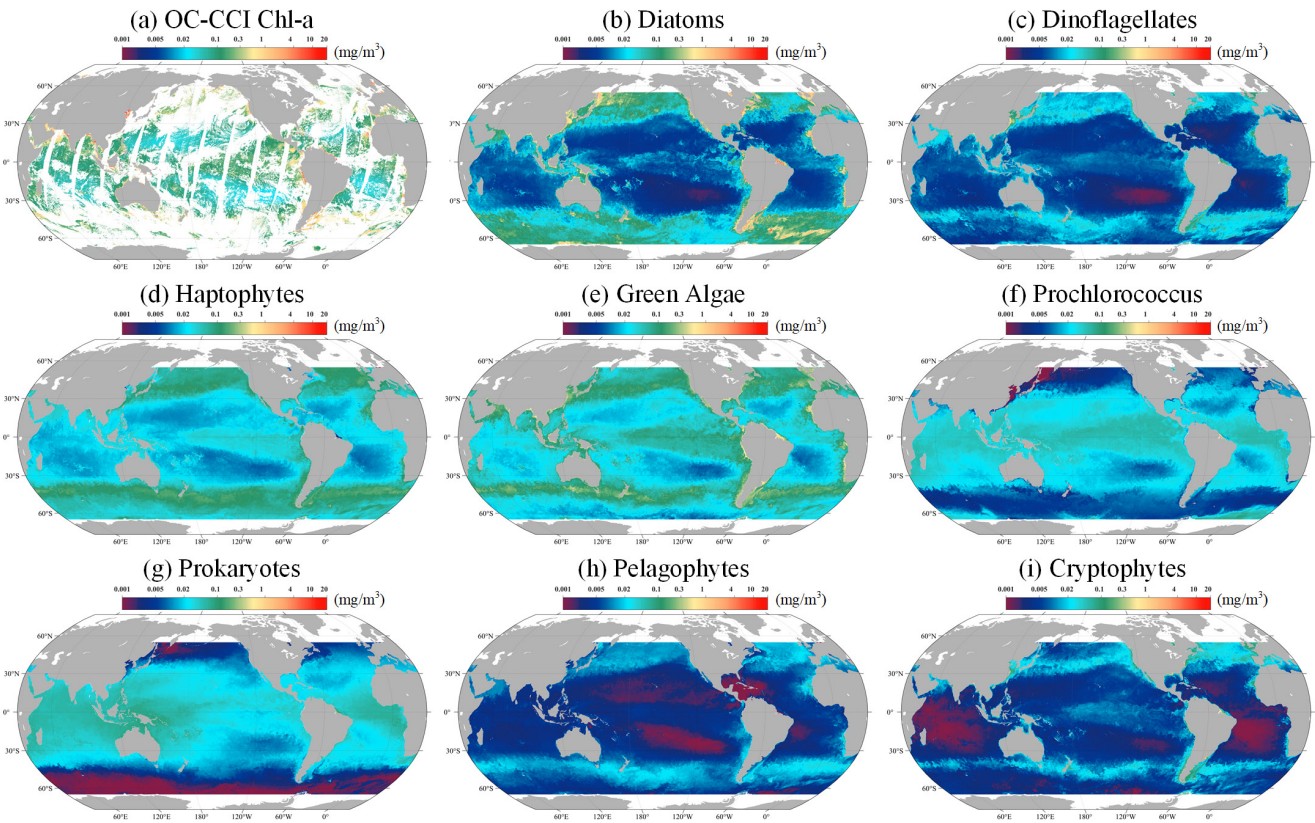

**Figure 12** The global distribution (2020-03-10) of the Chl-a concentration for (a) original OC-CCI, (b) Diatoms, (c) Dinoflagellates, (d)
Haptophytes, (e) Green Algae, (f) Prochlorococcus, (g) Prokaryotes, (h) Pelagophytes and (i) Cryptophytes. The grey areas represent lands.

Figure 13 delineated the corresponding uncertainties. Overall, the uncertainty relatively low in the open ocean, suggesting that
the model performs with a high degree of confidence. However, in coastal regions such as the East China Sea and the Amazon
River estuary, uncertainties escalate. This increase likely results from the complex coastal processes and land-sea interactions
prevalent in these areas, which can significantly influence the distribution and concentrations of PFTs, thereby challenging the
model's predictive accuracy. Despite the coastal uncertainties, Figure 13 also reveals that AIGD-PFT maintains globally low
uncertainty levels (below 0.1) for Diatoms, Dinoflagellates, Haptophytes, and Prokaryotes, highlighting the model's overall
stability and reliability. Additionally, Prochlorococcus exhibits higher uncertainties in the Southern Ocean, while Cryptophytes
show increased uncertainty in the equatorial Pacific. The reasons for this specific pattern require further investigation.

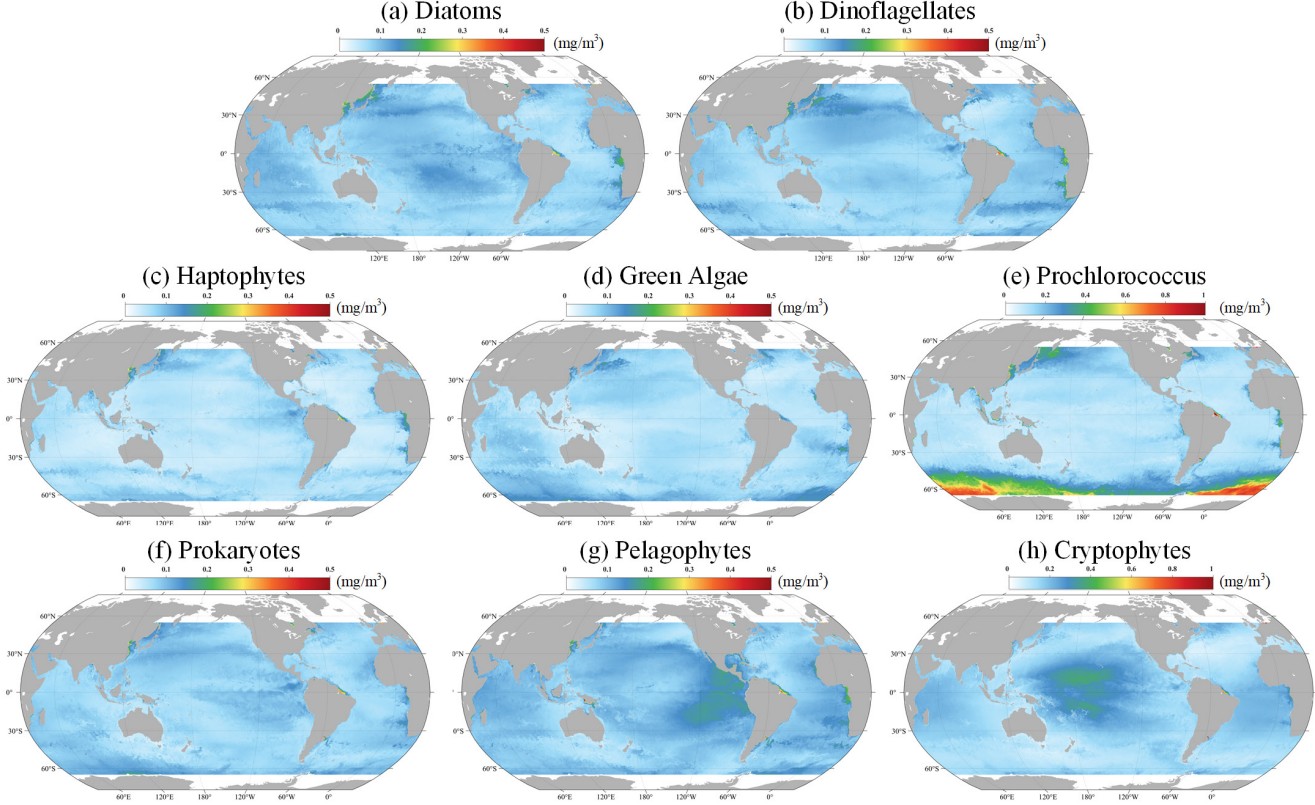


**Figure 13** The global distribution (2020-03-10) of the uncertainties for (a) Diatoms, (b) Dinoflagellates, (c) Haptophytes, (d) Green Algae, (e) Prochlorococcus, (f) Prokaryotes, (g) Pelagophytes and (h) Cryptophytes.

Further, Figure 14 illustrated the AIGD-PFT's ability to capture dynamic coastal processes, such as estuary runoff and coastal circulations, through time-series images of Diatom distribution in the Amazon River estuary (Figure 11a) and the Gulf of

Mexico (Figure 11b). The high Diatom concentrations near the Amazon River estuary, as shown in Figure 6a, correlated with the area's rich nutrient influx, also capturing the influence of the North Brazil Current (NBC) along the Brazilian coastline on Diatom dispersion. Figure 6b demonstrated the AIGD-PFT's efficacy in depicting the characteristics dominated by circulation and associated eddies in the Gulf of Mexico.



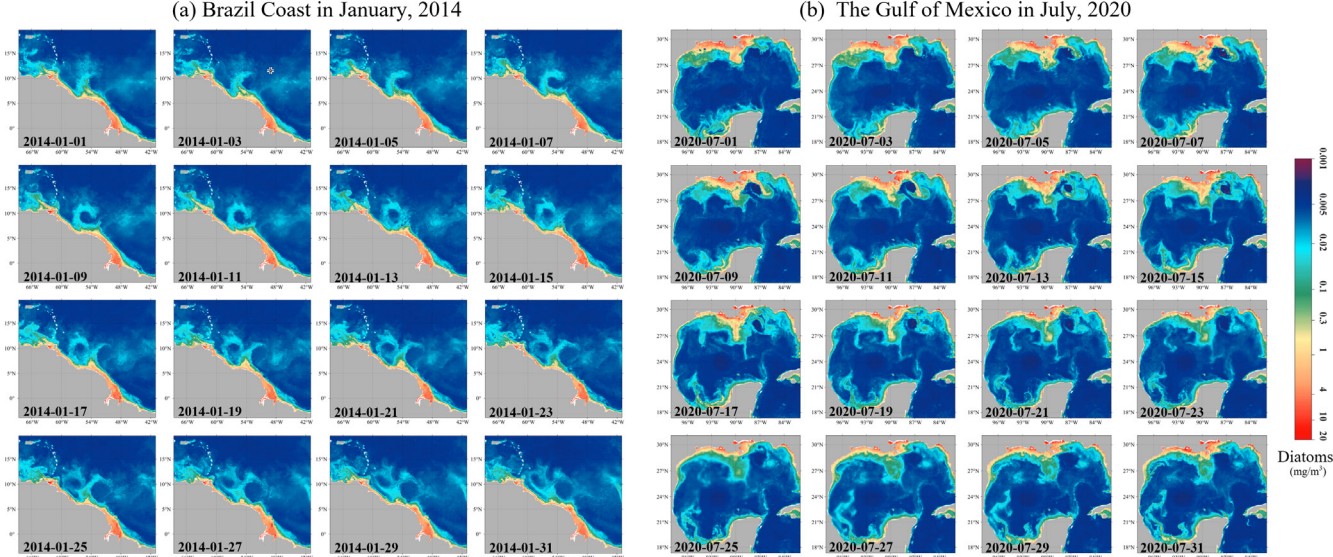

**Figure 14** Gap-free Diatoms in (a) Brazil Coast in January, 2014 and (b) Gulf of Mexico in July, 2020.

### 3.3 TCA-based Assessment

As depicted in Figure 15, we conducted a TCA on three daily-scaled PFT products: AIGD-PFT, SynSenPFT, and NOBM-daily. Figure 15a presents the statistical analysis results of correlation coefficients ($R$) and mean square error ($fMSE$) on a global scale. Meanwhile, Figure 15b, Figure 15c, and Figure 15d detail the comparative assessment results across different marine regions. Globally, the AIGD-PFT product outperforms the other two, demonstrating the highest median correlation values with actual conditions for Diatoms (0.81), Haptophytes (0.80), and Prokaryotes (0.72), respectively. AIGD-PFT product also have the lowest $fMSE$ values for all three PFTs, confirming its superiority with values of 0.35, 0.35, and 0.48, respectively. Comparatively, the SynSenPFT product underperforms relative to NOBM-daily in estimating Diatoms and Prokaryotes, yet excels in estimating Haptophytes.

The regional analysis (Figure 15b, 15c, and 15d) reveals variation in $R$ and $fMSE$ values across regions and PFTs. AIGD-PFT consistently outperforms in most regions for Diatom estimation but shows a slight increase in $fMSE$ in the equatorial Pacific, indicating a potential dip in estimation accuracy in this area. In contrast, SynSenPFT registers higher $fMSE$ values for Haptophytes estimation, particularly in the subtropical and southern Pacific regions. NOBM-PFT, on the other hand, tends to have lower correlation in Haptophytes estimation across regions, with a notable deficiency near the equatorial Pacific. Additionally, SynSenPFT demonstrates higher global $fMSE$ values for Prokaryotes compared to the other datasets, and NOBM-PFT significantly underperforms in Prokaryotes estimation in the Southern Ocean.



**Figure 15** TCA result of three daily product (AIGD-PFT, SynSenPFT, and NOBM-daily).

Further extending our analysis to monthly products (AIGD-PFT, EOF-PFT, NOBM-monthly), detailed in Figure 16. We
observed that AIGD-PFT and EOF-PFT exhibit closely matched performances for Diatoms, with median R values of 0.87 and
0.86, and *fMSE* of 0.24 and 0.25, respectively. Their Cumulative Distribution Function (CDF) curves nearly align perfectly.
Although global assessments for Diatoms are consistent, regional discrepancies exist. For instance, AIGD-PFT and EOF-PFT
perform similarly in the subtropical Pacific and the Indian Ocean, but AIGD-PFT achieves superior correlation in the equatorial
Pacific, Southern Ocean, and subtropical Atlantic. Conversely, EOF-PFT performs better in the South Pacific and equatorial
Atlantic. For Haptophytes and Prokaryotes, in summary, both global and regional assessments suggest that AIGD-PFT is the
most effective dataset, offering the lowest median *fMSE* and highest median *R* values. It stands out not only on a global scale
but also in most regional evaluations, confirming its overall superiority among the comparative datasets.

Earth System
Science
Data



**Figure 16** TCA result of three monthly product (AIGD-PFT, EOF-PFT, and NOBM-monthly).

## 4 Discussion

Phytoplankton serves as the foundation of marine food chains. Comprehensive monitoring and inversion of the spatiotemporal distribution patterns of Photosynthetic Functional Types (PFTs) are crucial for a deeper understanding of marine ecosystem functions, predicting and mitigating climate change, and other aspects. Amidst increasing human reliance on marine resources, maintaining the sustainability of fisheries and ensuring the stability and health of marine, especially coastal, ecosystems have become particularly urgent. This necessitates higher quality and more detailed phytoplankton diversity data to assist decision-making. However, existing satellite PFT products have significant shortcomings in inversion accuracy, spatiotemporal resolution, spatial coverage, and temporal span, limiting their application in climate and ocean management research.



Therefore, enhancing the quality and coverage of PFT data, with higher temporal resolution, is essential to reveal the immediate impacts of environmental changes on PFT distribution. Improved spatial coverage would enable more accurate descriptions of

local changes in marine ecosystems, providing more precise data support for scientific management strategies. Additionally, extending the temporal span would enhance the accuracy of long-term trend analysis, thereby better understanding the evolution of marine ecosystems.

Multi-source marine big data exhibits complementary advantages in terms of spatial integrity and accuracy. By merging data from various environmental factors, we can produce improved PFT products. In this study, we selected features including

ocean color data, biogeochemistry, temperature and salinity, and spatiotemporal information. Among these, ocean color data, as a crucial predictor, was seamlessly reconstructed using a GPU-accelerated DCT-PLS algorithm, filling gaps caused by clouds, orbits, and other factors. Compared to traditional reconstruction algorithms, the DCT-PLS algorithm is faster and effectively addresses the issue of missing observational data, improving data utilization efficiency and monitoring continuity.

Further, by leveraging the powerful nonlinear modelling capabilities of deep learning, we enhanced the accuracy of PFT

inversion. We developed a spatiotemporal ecological integration model based on deep learning, adapting the method proposed by Zhang et al. (2023) for reconstructing global PFTs from 1998 to 2023. The model, composed of 100 ResNet network models, demonstrates strong nonlinear modelling capabilities and robustness. Using the Monte Carlo method, we utilized ensemble means and standard deviations as the optimal estimates and uncertainties, generating a temporally continuous global PFT product covering the entire period and the corresponding uncertainty fields. The standard deviation reflects the variability of

model predictions, indicating the consistency between model predictions, i.e., the level of uncertainty.

We also employed three cross-validation methods to comprehensively validate the accuracy. Standard five-fold cross-validation focuses on the model's performance across the entire dataset, time-block five-fold cross-validation assesses the model's handling of time series, and space-block five-fold cross-validation concentrates on the model's ability to capture spatial distribution patterns. The results show that the STEE model generally exhibits good accuracy, demonstrating excellent

performance and stability in addressing temporal and spatial generalization issues. Notably, the model's high adaptability to reconstructed pixels highlights its potential for handling incomplete or inaccurate data, further proving the effectiveness of integrating ecological parameters and machine learning techniques. By applying the STEE model to all data from 1998 to 2023, we achieved accurate and robust monitoring of global high-resolution, spatiotemporally continuous PFT products. The TCA algorithm was used to compare the AIGD-PFT product with other products, showing that our estimation model achieved

competitive overall accuracy.

Despite statistical and correlational analyses throughout the paper confirming the reasonable and reliable estimation of global PFTs by STEE-DL, some uncertainties and limitations still need to be addressed in further work. Firstly, the variance obtained through ensemble learning mainly focuses on model prediction variability, but this does not fully capture or explain the actual



product uncertainties. Real product uncertainties are broader, encompassing incompleteness of actual measurements, uncertainties in predictors, and limitations in understanding the system. Exploring more comprehensive and precise uncertainty estimation methods to further enhance model reliability and applicability is necessary. Additionally, the current STEE-DL model is solely based on statistical relationships, lacking simulation of biological processes and therefore unable to explain mechanisms behind phytoplankton abundance changes. Model interpretability will be a focus of our future work. Incorporating prior information constraints such as ecological principles, biogeographical distributions, and seasonal changes into the model, constructing physics-guided neural networks, or achieving a symbiotic integration of physical methods and artificial intelligence, will create models that can accurately predict phytoplankton abundance with high interpretability.

The AIGD-PFT product demonstrates the potential application of artificial intelligence and marine big data in PFT modelling. This study focuses on the production process and product verification of AIGD-PFT, and a deeper analysis of PFT variations across different spatial and temporal dimensions will be the next research priority. As the product with the longest current time span (1998-2023) and continuous space-time coverage, AIGD-PFT has the potential to avoid false multi-year fluctuations and trend artifacts caused by data gaps. It helps in understanding the global and local trends of PFTs more broadly and is likely to reveal how climate change affects the composition of phytoplankton. This is crucial for predicting changes in marine ecosystems in the future, assessing the impact of climate change on the marine carbon cycle, and formulating corresponding conservation and management measures.

## 5 Data Availability

The AIGD-PFT (1998-2023, daily) dataset is stored in NetCDF format and can be accessed directly through: https://doi.org/10.11888/RemoteSen.tpdc.301164 (Zhang and Shen, 2024a). A video demonstration is available at https://doi.org/10.5446/67366 (Zhang and Shen, 2024b). In addition, a subset of AIGD-PFT (January 2023) can be downloaded at: https://doi.org/10.5281/zenodo.10910206 (Zhang and Shen, 2024c).

## 6 Conclusions

Constructing long time series models of global Photosynthetic Functional Types (PFTs) has always been a challenging task, with existing PFT products facing a variety of issues. To refine the monitoring of global phytoplankton groups, this study developed a deep learning-based spatiotemporal ecological integration model by combining multi-source marine data and artificial intelligence technology. This model can utilize a wide range of data sources, including ocean color, reanalysis, and in situ observations, to retrieve and generate the world's first daily updated, 4km resolution seamless PFT product, covering eight major phytoplankton groups. Cross-validation accuracy assessments show that our method can provide accurate and temporally consistent PFT predictions, demonstrating good performance in TCA evaluations across different products. As the first phytoplankton group product covering a 26-year span on a daily basis, the AIGD-PFT data aids in analyzing trends and interannual variations in phytoplankton time series, with the potential to reveal mechanisms by which phytoplankton

compositions respond to climate change across multiple time and spatial scales. Additionally, the AIGD-PFT product can facilitate the quantification of marine carbon fluxes and improve the accuracy of biogeochemical models. By deepening our understanding of these key components of marine ecosystems, we can more effectively address the challenges posed by climate change, ensuring the health of global ecosystems and the sustainable development of human society.

**Author contributions**

Conceptualization – Project Administration: Yuan Zhang, Fang Shen;

Methodology: Yuan Zhang, Fang Shen, Renhu Li, Mengyu Li, Zhaoxin Li;

Writing – Original: Yuan Zhang;

Writing – Review & Editing: Yuan Zhang, Fang Shen, Renhu Li, Mengyu Li, Zhaoxin Li, Songyu Chen, Xuerong Sun.

**Competing interests**

The authors declare that they have no known competing financial interests or personal relationships that could have appeared to influence the work reported in this paper.

**Disclaimer**

Publisher's note: Copernicus Publications remains neutral with regard to jurisdictional claims in published maps and institutional affiliations.

**Acknowledgements**

This study has made use of various in situ observations and databases, and the authors would like to thank the many scientists and crew involved with collecting and processing these data and making them freely and publicly available. All data used are properly cited and referred to in the reference list.

**Financial support**

This study was funded by the National Natural Science Foundation of China [Grant No. 42076187 and No.42271348].

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
