# Peer review of "AIGD-PFT: The first AI-driven Global Daily gap-free 4 km Phytoplankton Functional Type data product from 1998 to 2023"

_Earth System Science Data, 2024_

## Author Comment (AC1)

**Detailed Responses**

Here, we provide detailed responses to the referee #1' comments. The comments from the referees are shown in black. Our responses to the critics of the referees are supplied in normal font and **blue**. The appropriate correction in the manuscript has been repeated in **red** font in the response letter.

**Referee #1:**

General comments:

The paper by Zhang et al. presents the first AI-driven product for Phytoplankton Function Types (PFT) for the global ocean (AIGD-PFT). The AIGD-PFT consists of a L4 gap-free product including 8 PFT at daily and 4-km resolution for the period 1998-2023. AIGD-PFT is generated using an extended ensemble modelling approach (STEE-DL), which is based on machine and deep learning technologies and includes 100 models. Each model is built on statistical relationships between the physical environment and phytoplankton community and incorporates in situ HPLC data, ocean colour satellite observations whose missing data have been reconstructed throughout a cost-efficient DCT-PLS method, physical data from reanalysis and biogeochemical inputs from hindcast simulations.

Overall, the study falls within the scope of ESDD, methods are robust, and the manuscript is well written and detailed. Moreover, I believe that the AIGD-PFT product will be a very useful tool for all scientists interested in detecting climate-induced changes in the phytoplankton community. Therefore, I recommend this paper for publication, although I feel that some clarifications should be addressed to strengthen the way it is presented.

**Response:**

We are very grateful for reviewing our manuscript and providing us with your recognition and valuable advices on our work. Your comments and suggestions will definitely help us improve the manuscript.

We have revised the manuscript according to your specific comments and improved the quality. Please check the flowing item-by-item response, as well as the revised manuscript. Note that the appropriate corrections in the manuscript have been repeated in red font in the response letter.

Specific comments:

Authors present the AIGD-PFT as the product with the longest time span, covering 26 years (i.e., 1998-2023). However, I double checked the data sets used to create it and found some discrepancies that need to be clarified. In particular, except for the ESA-OC-CCI data set, which covers the whole period, I found that SST data from https://doi.org/10.48670/moi-00169 and biogeochemical variables from https://doi.org/10.48670/moi-00019 are available until October 2022 and December 2022, respectively, while SSS from https://doi.org/10.48670/moi-00016 is available from January 2022 to June 2024. So, I am not sure how authors create a 26-year product using some data sets that do not cover the same period.

**Response:**

Thank you for your detailed review. We apologize for the errors and confusion in our manuscript. We would like to clarify the specifics of the data used in our research as follows:

(1) **Sea Surface Temperature (SST) Data**: For SST, we utilized data from the ESA SST CCI and C3S reprocessed sea surface temperature analyses (DOI: https://doi.org/10.48670/moi-00169) which covers up to October 2022. For the period from November 2022 onwards, we employed the Global Ocean OSTIA Sea Surface Temperature and Sea Ice Analysis (DOI: https://doi.org/10.48670/moi-00165).

(2) **Sea Surface Salinity (SSS) Data**: We utilized the dataset Global Ocean Physics Reanalysis for SSS data (DOI: https://doi.org/10.48670/moi-00021, Fig. #1-1). This dataset includes the subset cmems_mod_glo_phy_my_0.083deg_P1D-m covering data before June 2021, and the subset cmems_mod_glo_phy_myint_0.083deg_P1D-mcovering from June 2021 onwards.

[Figure]

Fig. #1-1 Global Ocean Physics Reanalysis for SSS data. (DOI: https://doi.org/10.48670/moi-00021)

(3) **Biogeochemical Variables**: Regarding the biogeochemical variables, we used the Global Ocean Biogeochemistry Hindcast dataset (DOI: https://doi.org/10.48670/moi-00019, Fig. #1-2), which consists of two subsets. Until December 2022, we used the subset cmems_mod_glo_bgc_my_0.25deg_P1D-m, and from January 2023 onwards, we employed the subset cmems_mod_glo_bgc_myint_0.25deg_P1D-m.

[Figure]

Fig. #1-2 Global Ocean Biogeochemistry Hindcast dataset. (DOI: https://doi.org/10.48670/moi-00019)

We have added a clear statement (see line 198-206 on page 11 of revised manuscript), as follows:

"The SST data are sourced from the ESA SST CCI (Climate Change Initiative) and C3S (Copernicus Climate Change Service) global Sea Surface Temperature Reprocessed product (https://doi.org/10.48670/moi-00169, covering the period from January 1998 to October 2022) and Global Ocean OSTIA Sea Surface Temperature and Sea Ice Analysis (https://doi.org/10.48670/moi-00165, covering the period from November 2022 to December 2023). The SSS data are obtained from Global Ocean Physics Reanalysis (https://doi.org/10.48670/moi-00021). Biogeochemical data include nitrate concentration (NC), phosphate concentration (PC), silicate concentration (SC), and dissolved oxygen (DO). These variables are critical for understanding the nutrient dynamics in marine ecosystems, which are fundamental factors influencing phytoplankton growth and distribution. The data for these biogeochemical variables are sourced from the global biogeochemical multi-year hindcast products (https://doi.org/10.48670/moi-00019)."

2) As reported in Sect. 2.2.3, all physical and biogeochemical data have been resampled to a 4 km resolution, and I believe that this was done to match the high spatial resolution

of the ESA-OC-CCI product. However, any time data are resampled to a higher resolution, a greater but false accuracy is introduced due to the assumption that all new pixels have the same value when it may only be true for one pixel. This is why, as far as I know, the remapping direction is typically from high to low resolution. I would therefore ask authors to discuss this choice and, if possible, include a reference to previous works applying the same strategy. An interesting paper that may help the discussion can be found at https://journals.ametsoc.org/view/journals/apme/60/11/JAMC-D-20-0259.1.xml.

**Response:**

Thank you for pointing out this important concern.

We agree with you. As demonstrated in the study by Rajulapati et al. (2021) that you recommended, resampling from a lower to a higher resolution indeed can alter the statistical properties of the data, thereby introducing potential inaccuracies. In our study, we opted to resample all physical and biogeochemical data to the same high 4 km resolution as the ESA-OC-CCI product primarily for consistency across datasets. We acknowledge that transforming data from a lower to a higher resolution often assumes that the newly generated pixel values are similar to the original ones, potentially introducing a so-called "false precision" that could lead to systematic biases.

To minimize the impact of false precision, the Inverse Distance Weighting (IDW) method was employed for spatial interpolation. The IDW identifies all available pixels around a target pixel based on a search radius of 8 pixels, and the weights of the identified available pixels are then calculated by the reciprocal of the square of the distance between the target pixel and the available pixels. This method is more likely to provide balanced estimates and reduce the risk of introducing false precision.

With advancements in technology, the availability of high-resolution ocean data is increasing, such as *Multi-Scale Ultra High Resolution (MUR) Sea Surface Temperature data* (1km resolution, DOI: https://doi.org/10.5067/GHGMR-4FJ04), which provides hope for fundamentally addressing these issues. However, at present, offering datasets with varying spatial and temporal resolutions seems impractical. The resampling approach we have taken is a compromise intended to maximize the use of existing data resources while minimizing the computational and data processing burden. How to reduce information loss during data processing will be an important focus for our future work.

Rajulapati, C. R., Papalexiou, S. M., Clark, M. P., and Pomeroy, J. W.: The Perils of Regridding: Examples Using a

Global Precipitation Dataset, J Appl Meteorol Clim, 60, 1561-1573, https://doi.org/10.1175/Jamc-D-20-0259.1, 2021.

Follow your concerns, we have added a clear explanation about resampling (see line 206-212 on page 11 of revised manuscript):

"All data undergo the following preprocessing steps: (1) resampling, where all data is resampled to a 4km resolution using the pysample library (https://doi.org/10.5281/zenodo.3372769). The Inverse Distance Weighting (IDW) method was employed for spatial interpolation. The IDW identifies all available pixels around a target pixel based on a search radius of 8 pixels, and the weights of the identified available pixels are then calculated by the reciprocal of the square of the distance between the target pixel and the available pixels. This resampling process may lead to missing pixels, which are then filled using the nearest neighbor method;"

Additionally, the Discussion section has been expanded to include the following content (see line 523-528 on page 30-31 of revised manuscript):

"Firstly, in this study, all physical and biogeochemical data were resampled to match the high resolution of 4 km, consistent with the OC-CCI product, primarily to ensure uniformity across datasets, and to maximize the use of existing data resources. However, resampling from a lower to a higher resolution can indeed alter the statistical properties of the data, potentially introducing inaccuracies. In future research, it is planned to incorporate more high-resolution data and to minimize the loss of information during the data processing stage."

3) Page 8, line 151: The sentence needs to be reworded because, as reported in the Product Guide (https://docs.pml.space/share/s/fzNSPb4aQaSDvO7xBNOCIw), the latest ESA-OC-CCI product (v6.0) also merges observations from OLCI-3A and OLCI-3B.

**Response:**

Thank you for your reminder. We rephrased the relevant text (see line 150 on page 8 of revised manuscript) as follows:

"This dataset is generated by band-shifting and bias-correcting SeaWiFS, MODIS, VIIRS, and Sentinel 3A and 3B OLCI data to match MERIS data, achieving a spatial

resolution of 4 km"

4) I found the method used by authors to fill OC data gaps well described in Sect. 2.2.2. However, I think that specifying the number of available data before and after the filling procedure would be interesting and emphasize the effort authors have made. This information could also be presented by replacing Figure 3 with two Hovmöller diagrams showing the number of observations before and after the filling as function of time and latitude.

**Response:**

Thank you for your suggestion.

We have revised Figure 3 to include specific information on the changes in the quantity of available data before and after the filling process. Additionally, we have introduced two Hovmöller diagrams to visually represent these changes over time and latitude.

[Figure]

Figure 3 (a) Percentage of valid pixels in the OC-CCI v6.0 daily dataset; Hovmöller diagrams of (b) original OC-CCI data and (c) data after gap filling using the DCT-PLS method; (d) Comparison of the number of valid pixels between reconstructed and original data.

5) The choice to include the 8 PFTs as listed in the manuscript should be justified. I think that adding reference(s) should be enough to do that.

**Response:**

Thank you for your suggestion. We have added the relevant references (see line 137 on page 7 of revised manuscript):

"By utilizing an updated Diagnostic Pigment Analysis (DPA) methodology, along with newly adjusted weighting coefficients, we conducted DPA to ascertain in-situ PFT Chl-a concentrations. This analysis includes eight major PFTs: Diatoms, Dinoflagellates, Haptophytes, Pelagophytes, Cryptophytes, Green Algae, Prokaryotes, and Prochlorococcus, following conventional practices in the field (Xi et al., 2020; Xi et al., 2021)."

Xi, H. Y., Losa, S. N., Mangin, A., Garnesson, P., Bretagnon, M., Demaria, J., Soppa, M. A., D'Andon, O. H. F., and Bracher, A.: Global Chlorophyll a Concentrations of Phytoplankton Functional Types With Detailed Uncertainty Assessment Using Multisensor Ocean Color and Sea Surface Temperature Satellite Products, J Geophys Res-Oceans, 126, e2020JC017127, https://doi.org/10.1029/2020JC017127, 2021.

Xi, H. Y., Losa, S. N., Mangin, A., Soppa, M. A., Garnesson, P., Demaria, J., Liu, Y. Y., D'Andon, O. H. F., and Bracher, A.: Global retrieval of phytoplankton functional types based on empirical orthogonal functions using CMEMS GlobColour merged products and further extension to OLCI data, Remote Sens Environ, 240, 111704,   https://doi.org/10.1016/j.rse.2020.111704, 2020.

6) The definition of ResNet models (i.e., residual neural networks) is given in Sect. 2.3.1, but I think it should be provided earlier as they are mentioned before Sect. 2.3.1.

**Response:**

Thank you for your suggestion. We have adjusted the definition of ResNet models, moving it to the first instance where the concept appears (see line 98-101 on page 4 of revised manuscript):

"Here, we propose a novel Spatial–Temporal–Ecological Ensemble model based on deep learning (STEE-DL), designed to produce a long time series PFT product. STEE-DL leverages an ensemble of 100 ResNet (residual neural networks) models, incorporating inputs from reconstructed missing ocean color data, physical reanalysis, biogeochemical, and spatiotemporal information."

7) I suggest authors to go through the manuscript and split some long sentences to make the text more readable. For example, the second sentence in the abstract, which starts on line 2 and ends on line 14, can be split into at least three sentences.

**Response:**

Thank you for your suggestion. We rephrased the relevant text (see line 8 on page 7 of revised manuscript) as follows:

"In this study, we integrated artificial intelligence (AI) technology with multi-source marine big data to develop a Spatial–Temporal–Ecological Ensemble model based on Deep Learning (STEE-DL). This model generated the first AI-driven Global Daily gap-free 4 km PFTs product from 1998 to 2023 (AIGD-PFT). The AIGD-PFT significantly enhances the accuracy and spatiotemporal coverage of quantifying eight major PFTs: Diatoms, Dinoflagellates, Haptophytes, Pelagophytes, Cryptophytes, Green Algae, Prokaryotes, and Prochlorococcus."

8) I found some errors in the reference list (e.g., Zhang and Shen, 2024a,b,c). Please, check them carefully against the references as cited in the abstract and main text.

**Response:**

Thank you for the reminder. We have corrected it.

To conclude, I would like to mention that, as stated by authors, model interpretability is beyond the scope of this manuscript and will be a focus of a future work. I look forward to that. So, keep up the good progress!

**Response:**

Thank you for your encouragement comments. We appreciate your support and are committed to making model interpretability a key focus of our future research.

---

## Author Comment (AC2)

**Detailed Responses**

Here, we provide detailed responses to the referees #2' comments. The comments from the referees are shown in black. Our responses to the critics of the referees are supplied in normal font and **blue**. The appropriate correction in the manuscript has been repeated in **red** font in the response letter.

**Referee #2:**

The manuscript and datasets submitted by Zhang et al. proposed a thorough scheme AIGD-PFT using deep learning techniques to retrieve seamlessly eight phytoplankton functional types (PFTs) chlorophyll a concentrations on the global scale. The AIGD-PFT is built based on an extensive global in situ pigment data set and CMEMS products including satellite ocean color, physical and biogeochemical data sets based on model simulations covering the year from 1998 to 2023. All CMEMS data were preprocessed to have the same spatial resolution. Before performing the deep learning ensemble for PFT retrievals, a gap-filling technique DCT-PLS was firstly applied to all the global CMEMS products to generate seamless data on the global scale. The STEE-DL model were trained and established based on ResNet models using Monte Carlo and bootstrapping methods to finally estimate the PFT chlorophyll a concentration with corresponding model uncertainty assessment. Products were intercompared with other PFT data based on different methods and model simulations and showed outstanding performance.

This work demonstrated thoroughly the seamless PFT products on the global scale over the last 26 years and has shown high potential of machine learning/deep learning techniques in ocean color applications, and here especially for PFT information retrievals. This study delivered the first gap-free global PFT products. I find it significant and the study has put a big step forward for the phytoplankton group estimation using multiple products based on big-data deep learning methods. However, I have several comments and suggestions (listed below) that the authors may consider to hopefully help improve further the quality of this work.

**Response:**

We are grateful for reviewing our manuscript and providing us with your

recognition and valuable advice on our work. Your comments and suggestions have helped us improve the manuscript.

Please check the flowing item-by-item response, as well as the revised manuscript and supplementary materials. Note that the appropriate corrections in the manuscript have been repeated in red font in the response letter.

Abstract: 'PFT values' here indicate PFT chlorophyll a concentration, correct? This should be clarified in the beginning and kept consistent through the whole ms.

**Response:**

Thank you for your reminder. Yes, 'PFT values' here indicate PFT chlorophyll a concentration. We have revised it as follows (see Lines 17 on page 1 of the revised manuscript) and ensured consistency throughout the manuscript:

"The STEE-DL model utilizes an ensemble strategy with 100 ResNet models, applying Monte Carlo and bootstrapping methods to estimate optimal PFT chlorophyll a concentration and assess model uncertainty through ensemble means and standard deviations."

L23-25 Have the time series and impact of climate change been reflected here? Otherwise it is not proper to put such statement here but can be more on a perspective tone.

**Response:**

Thank you for your comment.

In the paragraph (Lines 23-25) of the original manuscript, we aim to convey the potential applications of the AIGD-PFT product rather than present specific findings regarding the impacts of climate change. While the AIGD-PFT product provides comprehensive spatiotemporal data that aids in studying phytoplankton dynamics and their response to climate change, our current analysis does not directly quantify these impacts.

Based on your suggestion, we have removed the relevant statements.

Intro: L43: Put also reference for DPA, Vidussi et al. 2001

**Response:**

Thank you for your attention to detail. The reference has been correctly cited (see line 38-40 on page 2 of the revised manuscript).

"the separation of phytoplankton diagnostic pigments through High-Performance Liquid Chromatography (HPLC) with the assistance of Diagnostic pigment analysis (DPA, Vidussi et al. 2001) or CHEMTAX (Mackey et al., 1996) algorithms remains the most cost-effective and quality-controlled method to date (Swan et al., 2016)."

Vidussi, F., Claustre, H., Manca, B. B., Luchetta, A., and Marty, J. C.: Phytoplankton pigment distribution in relation to upper thermocline circulation in the eastern Mediterranean Sea during winter, J Geophys Res-Oceans, 106, 19939-19956, https://doi.org/10.1029/1999jc000308, 2001.

L54-55: I think there are a few more references in this regard, e.g. El Hourany et al 2024, Li et al 2023 deep learning for pigments

**Response:**

Thank you. These references have been correctly cited (see line 50 on page 2 of the revised manuscript):

"…introducing more marine environmental covariates into ecological approaches (Zhang et al., 2023; Raitsos et al., 2008; El Hourany et al. 2024; Li et al. 2023) has become a current research focus…"

El Hourany, R., Karlusich, J.P., Zinger, L., Loisel, H., Levy, M., & Bowler, C. (2024). Linking satellites to genes with machine learning to estimate phytoplankton community structure from space. Ocean Science, 20, 217-239

Li, X.L., Yang, Y., Ishizaka, J., & Li, X.F. (2023). Global estimation of phytoplankton pigment concentrations from satellite data using a deep-learning-based model. Remote Sensing of Environment, 294

Sect 2.2.1 Indicate how many data were finally collected from all these sources

**Response:**

Thank you for your suggestion.

We have added Table S1 in the supplementary that details the number of data collected from each source:

Table S1 Reference and website for the publicly available in situ HPLC phytoplankton pigment dataset utilized in this study.

| No. | Coverage | Period | Number | Website |
|-----|----------|--------|--------|---------|
| 1 | Global | Aug 2000 – Apr 2018 | 4481 | https://doi.pangaea.de/10.1594/PANGAEA.938703 |
| 2 | South Atlantic Ocean | Nov 2000 – Mar 2012 | 2173 | https://doi.pangaea.de/10.1594/PANGAEA.848591 |
| 3 | Global | Nov 2004 – Sep 2012 | 146 | https://doi.pangaea.de/10.1594/PANGAEA.937536 |
| 4 | Global | Jul 2002 – Feb 2012 | 484 | https://doi.pangaea.de/10.1594/PANGAEA.930087 |
| 5 | Global | Dec 1988 – Aug 2012 | 15216 | https://doi.pangaea.de/10.1594/PANGAEA.875879 |
| 6 | Kuroshio region | Oct 2009 | 206 | https://doi.pangaea.de/10.1594/PANGAEA.819108 |
| 7 | Peruvian upwelling zone | Dec 2012 | 239 | https://doi.pangaea.de/10.1594/PANGAEA.864786 |
| 8 | Fram Strait | Jul 2017 – Aug 2017 | 534 | https://doi.pangaea.de/10.1594/PANGAEA.894860 |
| 9 | Australian Waters | Dec 1997 – present | 6951 | https://portal.aodn.org.au/search?uuid=97b9fe73-ee44-437f-b2ae-5b8613f81042 |
| 10 | Eastern China seas | 2015-2022 | 405 | - |

L161 DINEOF – I think the original studies should be cited here too.

**Response:**

Thank you for your reminder. The reference has been correctly cited (see line 161 on page 9 of the revised manuscript).

"Previous studies have developed various methods for reconstructing missing pixels in remote sensing data, such as DINEOF (Data Interpolation Empirical Orthogonal Function) (Alvera-Azcárate et al., 2011; Liu and Wang, 2022),"

Alvera-Azcárate, A., Barth, A., Sirjacobs, D., Lenartz, F., and Beckers, J. M.: Data Interpolating Empirical Orthogonal Functions (DINEOF): a tool for geophysical data analyses, Mediterr Mar Sci, 12, 5-11, 2011.

L 175 Normalisation: the dataset is standardized by dividing by the spatial mean, for each day or all 30 days together?

**Response:**

Thank you for your comments. In this study, the data normalization process is as follows: we first calculate the spatial mean for the entire dataset (from 1998 to 2023). Then, we standardize the data for each day by dividing it by this long-term mean.

The relevant text has been revised for clarity (see line 175 on page 10 of the revised manuscript):

"(2) Normalization: To minimize differences in dimensions and magnitudes of data across different spatial regions, the dataset is standardized by dividing by the spatial mean. The spatial mean is calculated from the entire dataset spanning from 1998 to 2023."

L186-189: high missing values – not proper, high missing rates?

**Response:**

Apologies for the confusion.

The term "high missing values" should indeed be more accurately stated as "high missing rates". We have corrected it (see line 190 on page 10 of the revised manuscript).

"It is important to note that in areas of high latitude with extremely high missing rates"

Seems that the authors have cut the data based on latitudes as there is a straight cutoff

in the maps?

**Response:**

Thank you for your comments.

We performed data cropping to minimize the common problems of missing data and low reliability at high latitudes. This method has been applied in previous research, and we have cited the relevant literature in our paper.

L195: Remove the '.' or use comma after Table 2.

**Response:**

Thank you for your reminder. We have removed it.

L198-199: SSS – This CMEMS product contains data from 2019 to 2024 only. I suppose you used the physical analysis hindcast too. Should be both cited.

**Response:**

We apologize for the errors and confusion in our manuscript. We would like to clarify the specifics of the data used in our research as follows:

(1) **Sea Surface Temperature (SST) Data**: For SST, we utilized data from the ESA SST CCI and C3S reprocessed sea surface temperature analyses (DOI: https://doi.org/10.48670/moi-00169) which covers up to October 2022. For the period from November 2022 onwards, we employed the Global Ocean OSTIA Sea Surface Temperature and Sea Ice Analysis (DOI: https://doi.org/10.48670/moi-00165).

(2) **Sea Surface Salinity (SSS) Data**: We utilized the dataset Global Ocean Physics Reanalysis for SSS data (DOI: https://doi.org/10.48670/moi-00021, Fig. #1-1). This dataset includes the subset cmems_mod_glo_phy_my_0.083deg_P1D-m covering data before June 2021, and the subset cmems_mod_glo_phy_myint_0.083deg_P1D-mcovering from June 2021 onwards.

(3) **Biogeochemical Variables**: Regarding the biogeochemical variables, we used the Global Ocean Biogeochemistry Hindcast dataset (DOI: https://doi.org/10.48670/moi-00019, Fig. #1-2), which consists of two subsets. Until December 2022, we used the subset cmems_mod_glo_bgc_my_0.25deg_P1D-m, and

from January 2023 onwards, we employed the subset cmems_mod_glo_bgc_myint_0.25deg_P1D-m.

We have added a clear statement (see line 198-207 on page 11 of revised manuscript), as follows:

"The SST data are sourced from the ESA SST CCI (Climate Change Initiative) and C3S (Copernicus Climate Change Service) global Sea Surface Temperature Reprocessed product (https://doi.org/10.48670/moi-00169, covering the period from January 1998 to October 2022) and Global Ocean OSTIA Sea Surface Temperature and Sea Ice Analysis (https://doi.org/10.48670/moi-00165, covering the period from November 2022 to December 2023). The SSS data are obtained from Global Ocean Physics Reanalysis (https://doi.org/10.48670/moi-00021). Biogeochemical data include nitrate concentration (NC), phosphate concentration (PC), silicate concentration (SC), and dissolved oxygen (DO). These variables are critical for understanding the nutrient dynamics in marine ecosystems, which are fundamental factors influencing phytoplankton growth and distribution. The data for these biogeochemical variables are sourced from the global biogeochemical multi-year hindcast products (https://doi.org/10.48670/moi-00019)."

L205: Resampling from lower resolution to high res might cause irreal data filling

**Response:**

Thank you for pointing out this important concern.

We agree with you. In our study, we opted to resample all physical and biogeochemical data to the same high 4 km resolution as the ESA-OC-CCI product primarily for consistency across datasets. We acknowledge that transforming data from a lower to a higher resolution often assumes that the newly generated pixel values are similar to the original ones, potentially introducing a so-called "false precision" that could lead to systematic biases.

To minimize the impact of false precision, the Inverse Distance Weighting (IDW) method was employed for spatial interpolation. The IDW identifies all available pixels around a target pixel based on a search radius of 8 pixels, and the weights of the identified available pixels are then calculated by the reciprocal of the square of the distance between the target pixel and the available pixels. This method is more likely

to provide balanced estimates and reduce the risk of introducing false precision.

With advancements in technology, the availability of high-resolution ocean data is increasing, such as *Multi-Scale Ultra High Resolution (MUR) Sea Surface Temperature data* (1km resolution, DOI: https://doi.org/10.5067/GHGMR-4FJ04), which provides hope for fundamentally addressing these issues. However, at present, offering datasets with varying spatial and temporal resolutions seems impractical. The resampling approach we have taken is a compromise intended to maximize the use of existing data resources while minimizing the computational and data processing burden. How to reduce information loss during data processing will be an important focus for our future work.

Follow your concerns, we have added a clear explanation about resampling (see line 207-212 on page 11 of revised manuscript):

"All data undergo the following preprocessing steps: (1) resampling, where all data is resampled to a 4km resolution using the pysample library (https://doi.org/10.5281/zenodo.3372769). The Inverse Distance Weighting (IDW) method was employed for spatial interpolation. The IDW identifies all available pixels around a target pixel based on a search radius of 8 pixels, and the weights of the identified available pixels are then calculated by the reciprocal of the square of the distance between the target pixel and the available pixels. This resampling process may lead to missing pixels, which are then filled using the nearest neighbor method;"

Additionally, the Discussion section has been expanded to include the following content (see line 523-528 on page 30-31 of revised manuscript):

"Firstly, in this study, all physical and biogeochemical data were resampled to match the high resolution of 4 km, consistent with the OC-CCI product, primarily to ensure uniformity across datasets, and to maximize the use of existing data resources. However, resampling from a lower to a higher resolution can indeed alter the statistical properties of the data, potentially introducing inaccuracies. In future research, it is planned to incorporate more high-resolution data and to minimize the loss of information during the data processing stage."

Standardisation – is this step conflicting with the normalisation step 2 of the DCT-PLS?

**Response:**

Thank you for your comments. Although both standardization processes involve

data scaling and are technically similar, they serve two distinct purposes and operate independently within the processing workflow, with no conflict between them: (1) The normalization in DCT-PLS primarily aims at data reconstruction to ensure the completeness and continuity of the dataset; (2) The normalization used in the predictive model is designed to scale various input variables to a uniform level, thus enhancing the stability and effectiveness of model training.

L210-218: any basis/ references for these transformations?

**Response:**

Thank you for your comments. We have included relevant references in the revised manuscript to support the scientific basis for using these transformations (see line 216-217 on page 11 of revised manuscript):

"Incorporating spatial-temporal encoding into models is an effective strategy to enhance prediction accuracy, allowing for better capture of complex spatial-temporal interactions within the data (Yang et al., 2022; Wei et al., 2023)."

Wei, J., Li, Z. Q., Lyapustin, A., Wang, J., Dubovik, O., Schwartz, J., Sun, L., Li, C., Liu, S., and Zhu, T.: First close insight into global daily gapless 1 km PM pollution, variability, and health impact, Nat Commun, 14, https://doi.org/10.1038/s41467-023-43862-3, 2023.

Yang, N. S., Shi, H. Z., Tang, H., and Yang, X.: Geographical and temporal encoding for improving the estimation of PM concentrations in China using end-to-end gradient boosting, Remote Sens Environ, 269, https://doi.org/10.1016/j.rse.2021.112828, 2022.

L225 Is the STEE-DL model different from that in Zhang et al. 2023? Why did not the authors use that approach but developed the current STEE-DL instead? Any advantages?

**Response:**

Thank you for your comments. We responses to the above two questions one by one as follows:

(1) Is the STEE-DL model different from that in Zhang et al. 2023?

Yes, the proposed STEE-DL model in this study is different from the model described in Zhang et al. 2023. The previous STEE model from Zhang et al. (2023) combines three different machine learning methods (Gradient Boosting Machine, 1D-

CNN, and TabNet) using ridge regression for ensemble learning. In contrast, the STEE-DL model is built around an ensemble of 100 ResNet models.

(2) Why did not the authors use that approach but developed the current STEE-DL instead? Any advantages?

In the previous research by Zhang et al. (2023), the focus was primarily on the generation of monthly PFT products, for which the STEE model was developed. The STEE model integrates three complex machine learning methods aimed at achieving high prediction accuracy. However, when the present study shifted from monthly to daily predictions, the computational demand increased significantly, turning the processing speed of the model into a critical bottleneck. Additionally, although the previous STEE model is capable of making high-precision predictions, it does not provide an uncertainty assessment for these predictions, which is a drawback in many ecological applications.

These challenges prompted the development of the STEE-DL model. The proposed STEE-DL model represents a significant improvement and expansion over the STEE model described in Zhang et al. (2023). It features major enhancements in the following two areas:

(1) **Running Speed**: The STEE-DL model is entirely based on a deep learning architecture and was specifically designed with computational efficiency in mind. By employing lightweight network designs and leveraging GPU acceleration, it significantly reduces the time required for computations. This enhancement is particularly important for our study's application scenario, which involves processing massive datasets to generate global, long-term daily PFT (Plant Functional Type) data series.

(2) **Uncertainty Assessment**: The STEE-DL model incorporates a deep learning ensemble framework, which not only improves prediction accuracy but also enables the direct assessment of prediction uncertainty—a capability not present in the Zhang et al., 2023 model. By calculating the ensemble mean and standard deviation of the model outputs, the STEE-DL provides a quantified range of uncertainty for each prediction. This feature is extremely valuable for scientific research and decision-making support.

Zhang, Y., Shen, F., Sun, X. R., and Tan, K.: Marine big data-driven ensemble learning for estimating global phytoplankton group composition over two decades (1997-2020), Remote Sens Environ, 294, 113596, https://doi.org/10.1016/j.rse.2023.113596, 2023.

To clarify and follow your concerns, we have added a statement (see lines 229-236 on pages 12-13 of revised manuscript) as follows:

"In the previous research by Zhang et al. (2023), the focus was primarily on the generation of monthly PFT products, for which the STEE (Spatial-Temporal-Ecological Ensemble) model was developed. The STEE model integrates three complex machine learning methods aimed at achieving high prediction accuracy. However, when the present study shifted from monthly to daily predictions, the computational demand increased significantly, turning the processing speed of the model into a critical bottleneck. Additionally, although the previous STEE model is capable of making high-precision predictions, it does not provide an uncertainty assessment for these predictions, which is a drawback in many ecological applications. To address these challenges, the present study further developed the STEE-DL (Spatial–Temporal–Ecological Ensemble model based on deep learning)."

L232-233: reads strange. Rephrase the sentence - This setup decreases the dimensionality of features from 19 to 16, and then to 10, before a final fully connected layer maps these features to an output value for predicting the target variable.

**Response:**

We have rephrased the description to more clearly explain the model's process from input to output and the changes in network dimensions (see line 245-247 on page 13 of revised manuscript):

"In this model, the input layer receives 19 feature variables, which are then reduced to 16 after the first residual block. Subsequently, the second residual block further reduces the number of features to 10. Finally, a fully connected layer maps these features to an output value for predicting the target variable."

L245- put example references for statistical methods

**Response:**

Thank you for your suggestion. The relevant literature has been correctly cited as follows (see line 255 on pages 13 of revised manuscript):

"The variability among ensemble model outputs, quantified by the standard deviation ($\sigma$) of the 100 independent models, provides a measure of uncertainty in

predictions (Chau et al., 2022)."

Chau, T. T. T., Gehlen, M., and Chevallier, F.: A seamless ensemble-based reconstruction of surface ocean pCO and air-sea CO fluxes over the global coastal and open oceans, Biogeosciences, 19, 1087-1109, https://doi.org/10.5194/bg-19-1087-2022, 2022.

L253: Does this show how the matchups between the in situ data and CMEMS products were extracted? I would indicate the number of the data points too - also later in the stats

**Response:**

Thank you for your comments. In this study, the matchup data were generated based on temporal and spatial proximities. Specifically, we used a temporal window of one day and a spatial window of four kilometers for the matchups.

We have included Figure S1 in the supplementary material to illustrate the number of data points (see line 272 on pages 14 of revised manuscript):

"Figure S1 in the Supplementary material presents the histograms of the Chl-a concentrations of the eight PFTs at log-10 scale."

[Figure]

Figure S1 Log-scale histogram of Chl-a concentrations for eight PFTs and the number of in situ data points.

L288-L292: put this together this paragraph with the above one, or using bullets to describe the three CV procedures more clearly.

**Response:**

Thank you for your constructive feedback. Based on your suggestion, we have combined the mentioned sections and provided a clear description of the three cross-validation (CV) procedures. The revised section is as follows (see lines 293-309 on pages 15 of revised manuscript):

"Cross validation (CV) is a commonly used method for analyzing model performance, allowing for a comprehensive assessment of a model's accuracy, stability, and generalization. This study implements three types of CV methods: random five-fold CV, time-block five-fold CV, and spatial-block five-fold CV, to deeply evaluate the model's multifaceted performance. Specifically, the methods are as follows:

(1) Standard five-fold cross-validation: This method randomly divides all data into five equal-sized subsets. In each round of validation, one subset is selected as the test set, while the remaining four subsets serve as the training set, ensuring that each data point is used as test data. This method primarily evaluates the model's performance and generalization on the entire dataset.

(2) Time-block five-fold cross-validation: Data is divided into five consecutive time periods in chronological order. In each iteration, data from one time period is chosen as the test set, with the data from the remaining periods serving as the training set (as shown in Figure 5). This method takes into account the continuity and dependency of time series, helping to evaluate the model's ability to capture time trends and seasonal variations.

(3) Spatial-block five-fold cross-validation: Similar to time-block cross-validation, but data is divided based on spatial location. A hexagonal grid was created at 20° horizontal and vertical intervals, and regions without sampling points were removed for hexagonal regions. In each round, data from one geographical block is left out as the testset, while data from other blocks are used for training(as shown in Figure 6). This method prevents potential data leakage due to spatial autocorrelation and helps to assess the model's spatial prediction capability and its generalization across different geographical locations."

L386: not sure if it is appropriate to call them ecological types.

**Response:**

Thank you for your comments. we have replaced "ecological types" by "functional types" in the revised manuscript (see line 404 on pages 23 of revised manuscript), as follows:

"Firstly, the model achieved high prediction accuracy for key functional types such as Diatoms, Dinoflagellates, and Green algae, with significant advantages at certain sites: for instance, at sites 4 and 5, the prediction correlation coefficients for Diatoms were as high as 0.90 and 0.88, respectively."

L402-404: High missing rates in high latitudes limit the application there. Can the authors indicate the range of the latitudes for these seamless PFT products?

**Response:**

Thank you for your comments. Our PFT products are primarily applicable within the range of 75°S to 75°N. We have added a clear statement to indicate this range (see line 190 on pages 10 of revised manuscript), as follows:

"It is important to note that in areas of high latitude (above 75°) with extremely high missing values (exceeding 80%), these data will be directly removed (as demonstrated in the video example available at https://doi.org/10.5446/67366), because reconstruction under such conditions is impractical."

22) Fig 12: Though it is demonstrated in the video, maybe yearly mean maps here can better demonstrate the whole global ocean - a daily product cannot cover both polar regions.

**Response:**

Thank you for your suggestion. We have included yearly mean maps in the supplementary material of the revised manuscript (see line 432 on pages 25 of revised manuscript), as follows:

"Additionally, the yearly mean maps for 2020 are provided in Figure S2 of the supplementary, showing the distribution pattern of global ocean PFT throughout the year."

[Figure]

Figure S2 The yearly mean global distribution of Chl-a concentration in 2020 for (a) Diatoms, (b) Dinoflagellates, (c) Haptophytes, (d) Pelagophytes, (e) Cryptophytes, (f) Green Algae, (g) Prokaryotes and (h) Prochlorococcus. The grey areas represent lands.

Fig 13 and uncertainty: I see all data were log transformed, how were these uncertainties calculated in the original conc.?

**Response:**

We apologize for the mistake in Figure 13. Indeed, all computations of the

uncertainties in this study were conducted on logarithmically transformed data, which follows conventional practice in the field of ocean color research.

We have corrected Figure 13 and added a detailed explanation (see line 258 on pages 13 of revised manuscript), as follows:

"It should be noted that all computations of the uncertainties in this study were conducted on logarithmically transformed data, which follows conventional practice in the field of ocean color research (Xi et al., 2021)."

Xi, H. Y., Losa, S. N., Mangin, A., Garnesson, P., Bretagnon, M., Demaria, J., Soppa, M. A., D'Andon, O. H. F., and Bracher, A.: Global Chlorophyll a Concentrations of Phytoplankton Functional Types With Detailed Uncertainty Assessment Using Multisensor Ocean Color and Sea Surface Temperature Satellite Products, J Geophys Res-Oceans, 126, e2020JC017127, https://doi.org/10.1029/2020JC017127, 2021.

[Figure]

Figure 13 The global distribution (2020-03-10) of the uncertainties for (a) Diatoms, (b) Dinoflagellates, (c) Haptophytes, (d) Green Algae, (e) Prochlorococcus, (f) Prokaryotes, (g) Pelagophytes and (h) Cryptophytes.

L504-505: From Fig 13 the model uncertainties one can see already large uncertainties for certain PFT in some regions, such as diatoms and cryptophytes with very low chla values (<0.01 mg m-3) in the gyres but with uncertainty larger than 0.1 mg m-3 and also for Prochlorococcus in high latitudes (almost not existing) with very high uncertainty.

**Response:**

Thank you for your comments. We apologize once again for the unit error in Figure 13 and would like to clarify that the uncertainties were actually calculated on a logarithmic scale, not in mg m$^{-3}$, as previously corrected.

Regarding the high uncertainties in regions with low chlorophyll concentrations that you mentioned, there are two main reasons: (1) Insufficient training data. The lack of sufficient training data to represent these extreme conditions limits the model's ability to generalize the distribution of PFT in these areas, resulting in increased predictive uncertainty. (2) Intrinsic detection limits of the model. At very low chlorophyll concentrations, the sensitivity and accuracy of remote sensing technology can decrease, leading to a significant increase in relative uncertainty of predictions.

In future research, we are committed to improving the model's performance under these extreme conditions by incorporating more diverse and representative data.

L512 Discussion: How easy is it to apply the STEE-DL model to future datasets? I find it might be difficult to apply it as one has to prepare and preprocess all input data and fill the gaps using the DCT-PLS. That might be an obstacle to put it into operational. The authors should discuss on this point too.

**Response:**

Thank you for your reminder.

The STEE-DL model requires a relatively complex data preprocessing, including data cleaning, normalization, and filling gaps using the DCT-PLS method. Although these steps increase the workload before deploying the model, they are essential to ensure the quality and integrity of the model input data.

To reduce the complexity of these steps, we are developing more user-friendly data preprocessing tools, which will help users prepare data more easily.

Are the authors planning to publish the codes of AIGD-PFT in the future, so that the others can test it with their own prepared data sets?

**Response:**

Thank you for your comment. We plan to open source STEE-DL model and related tools in the future. This will include the complete data processing pipeline, model training, and prediction codes.

---

## Author Response (AR2)

**Detailed Responses**

Here, we provide detailed responses to the referee #2' comments. The comments from the referees are shown in black. Our responses to the critics of the referees are supplied in normal font and **blue**. The appropriate correction in the manuscript has been repeated in **red** font in the response letter.

**Referee #2:**

Thanks for the revised version. I have only a few corrections/comments and I hope the authors can incorporate them into the final version before acceptance.

**Response:**

We are very grateful for reviewing our manuscript and providing us with your recognition and valuable advices on our work. Your comments and suggestions will definitely help us improve the manuscript.

1. L200: It was noticed that the authors used SST CCI data before 2022 and OSTIA SST NRT products afterwards. Are the authors aware of the OSTIA reprocessed data https://doi.org/10.48670/moi-00168 covering 1988-2023, which I believe is a more consistent product with the OSTIA NRT data compared to the CCI SST. Have the authors checked on the difference of the OSTIA and CCI SST data? In some regions the difference can be significant.

**Response:**

Thank you for your detailed review.

Regarding the OSTIA reprocessed dataset you mentioned, we were indeed unaware of it in this study, and did not conduct an in-depth analysis of the differences between various SST products. Based on your suggestion, we performed a simple visual comparison and found that although there are differences between the OSTIA and CCI SST data, these differences are not significant at the spatiotemporal scales required for our research. Therefore, we believe that these differences have limited impact on the conclusions of our study.

Furthermore, we fully understand the importance of the variability in SST data for research accuracy. We deeply agree with your suggestion for further comparative analysis. However, considering the scope of this study, conducting a comprehensive

comparison would exceed the anticipated workload of this research. We will consider a more comprehensive comparative analysis of the OSTIA and CCI SST datasets in future studies to ensure the robustness and reliability of the results.

2. Terminology inconsistency: It was found both in the response letter and in the revised manuscript, PFT stands for different terms (e.g. Plant functional types in the response letter, and Photosynthetic functional types in the ms??). I believe the authors know what PFT really denotes. Please check throughout the text to make sure the consistency of all terms.

**Response:**

We sincerely apologize for the errors and any confusion caused by the inconsistent use of terminology in our manuscript and response letter.

PFT stands for *Phytoplankton Functional Type*. We have conducted a thorough review of the entire manuscript, ensuring that all references to PFT are now consistent and correctly denote *Phytoplankton Functional Type*.

3. It was mentioned how the predictor variables were preprocessed (normalised etc) but not for the in situ PFT data as response variables. How were they normalised before put into the ensemble training?

**Response:**

Thank you for your comment.

All PFT data are processed using the log-10 transformation, which is common in the field of ocean color research. This transformation helps to narrow the range of data, bringing the distribution closer to a normal distribution, thereby enhancing the stability and accuracy of model predictions.

4. Uncertainty is a bit unclear - are they based on log-10 or natural logarithmic transformed data? Can you explain how to understand it or how to convert it to the common relative errors (%)

**Response:**

Thank you for your detailed review of the uncertainty issue.

The uncertainty in our study is based on log-10 transformed data, which is a common approach in the ocean color field for analyzing bio-optical parameters. We have clarified this in the revised manuscript (see page 13, line 261).

"It should be noted that all computations of the uncertainties in this study were conducted on log-10 transformed data, which follows conventional practice in the field of ocean color research (Xi et al., 2021)."

Xi, H. Y., Losa, S. N., Mangin, A., Garnesson, P., Bretagnon, M., Demaria, J., Soppa, M. A., D'Andon, O. H. F., and Bracher, A.: Global Chlorophyll a Concentrations of Phytoplankton Functional Types With Detailed Uncertainty Assessment Using Multisensor Ocean Color and Sea Surface Temperature Satellite Products, J Geophys Res-Oceans, 126, e2020JC017127, https://doi.org/10.1029/2020JC017127, 2021.

It is important to clarify that the uncertainty described in this paper is not the typical relative error (i.e., the percentage deviation between predicted and true values). Instead, it is estimated through the variance of predictions from multiple sub-models in the ensemble learning process, as shown in the following figure.

[Figure]

Fig. 1 Schematic view of Ensemble Learning.

The uncertainty in our study is conceptually different from the relative errors (%), and thus direct conversion between the two is difficult. This approach does not rely on

simple point estimates but rather uses a distributional approach to characterize the dispersion of prediction results and model stability. In ensemble learning, uncertainty reflects the variability in model predictions when different training datasets or model architectures are used. This measure of uncertainty is more aligned with the requirements of evaluating ensemble model predictions in our study, as it better captures the model's robustness when faced with unseen data. Additionally, deep ensemble methods have been proven to be a Bayesian approach that can provide high-quality uncertainty estimates, outperforming methods like MC Dropout (Lakshminarayanan et al., 2017). Compared to traditional Bayesian methods such as Gaussian process regression, variational Bayesian, and Laplace approximation, ensemble learning offers significant advantages in flexibility, ease of implementation, and computational efficiency (Abdar et al., 2021). The concept of uncertainty in our study is different from the relative errors (%), and therefore cannot be directly converted.

Abdar, M., Pourpanah, F., Hussain, S., Rezazadegan, D., Liu, L., Ghavamzadeh, M., Fieguth, P., Cao, X. C., Khosravi, A., Acharya, U. R., Makarenkov, V., and Nahavandi, S.: A review of uncertainty quantification in deep learning: Techniques, applications and challenges, Inform Fusion, 76, 243-297, 10.1016/j.inffus.2021.05.008, 2021.

Lakshminarayanan, B., Pritzel, A., and Blundell, C.: Simple and Scalable Predictive Uncertainty Estimation using Deep Ensembles, Adv Neur In, 30, 2017.

5. Similar to my previous comment on Fig 12, Fig 13 doesn't show the uncertainty distribution in the north pole. Maybe put the maps of another month e.g. July or August in the Supplementary document.

**Response:**

Thank you for your comments. Follow your concerns, we have added the uncertainty distribution map for July 10, 2020, in the Supplementary to provide a more comprehensive view of the North Pole region.

"Additionally, Figure S3 in the supplementary materials illustrates the global distribution of uncertainties on July 10, 2020.

[Figure]

Figure S3 The global distribution (2020-07-10) of the uncertainties for (a) Diatoms, (b) Dinoflagellates, (c) Haptophytes, (d) Green Algae, (e) Prochlorococcus, (f) Prokaryotes, (g) Pelagophytes and (h) Cryptophytes."

6. L439 the uncertainty (is) relatively...

**Response:**

We apologize for the mistake. We have corrected it (see line 439 on page 25 of the revised manuscript).

"Overall, the uncertainty is relatively low in the open ocean, suggesting that the model performs with a high degree of confidence."

7. PFT maps - L430, Figs 12-13 Figure S2

Mind that: Prochlorococcus prediction in high latitudes is unrealistic as they are almost never found in oceans above 50 degrees if you have checked the global in situ HPLC data. Also see Flombaum et al 2013 and Xi et al. (2021). This should be minded and probably cut it off for the high latitudes and should also be discussed together with the uncertainty.

**Response:**

Thank you for pointing out this important concern.

We fully agree with you. As shown in Figure S3 of the supplementary materials, the uncertainty associated with Prochlorococcus predictions increases significantly at latitudes above 50°. This indicates that the predictions in these high-latitude regions may not be realistic. In future research and model improvements, we will consider introducing a threshold based on existing ecological studies and global in situ data analysis. This threshold will help filter out unrealistic predictions in high-latitude regions, particularly in areas with higher uncertainty, thereby achieving more accurate and ecologically consistent global PFT predictions. Additionally, incorporating more ecological prior knowledge as constraints during the neural network training process is crucial. This approach will not only help the model better understand and reflect ecological principles but also enhance its predictive capability and reliability in complex environments. We have added a brief discussion (see line 538 on page 31 of the revised manuscript).

"It is also necessary to consider introducing a threshold based on existing ecological studies and global in situ data analysis, which will help filter out predictions in areas with high uncertainty."

8. Regarding my last second comment about the difficulty of applying the STEE-DL model to future datasets - I suggested to have a brief discussion on this point in the revised version, however it is not found there.

**Response:**

Thank you for your comments. To facilitate the application of the STEE-DL model, we are developing a set of user-friendly data preprocessing tools that will help users

effectively utilize our STEE-DL model with updated datasets. Following your concerns, we have added a brief discussion (see line 502 on page 30 of the revised manuscript).

"As environmental data continues to be updated, the STEE-DL model can be easily applied to future datasets, allowing for the continuous generation of PFTs, which will contribute to long-term global or local scale analyses."